# MOMENTUM-ACCELERATED DIFFUSION PROCESS FOR FASTER TRAINING AND SAMPLING

## ABSTRACT

Diffusion models (DMs) have been adopted across diverse fields with its remarkable abilities in capturing intricate data distributions. In this paper, we propose a Fast Diffusion Model (FDM) to significantly speed up DMs from a stochastic optimization perspective for both faster training and sampling. We first find that the diffusion process of DMs accords with the stochastic optimization process of stochastic gradient descent (SGD) on a stochastic time-variant problem. Then, inspired by momentum SGD that uses both gradient and an extra momentum to achieve faster and more stable convergence than SGD, we integrate momentum into the diffusion process of DMs. This comes with a unique challenge of deriving the noise perturbation kernel from the momentum-based diffusion process. To this end, we frame the process as a Damped Oscillation system whose critically damped state—the kernel solution—avoids oscillation and yields a faster convergence speed of the diffusion process. Empirical results show that our FDM can be applied to several popular DM frameworks, *e.g.*, VP (Song et al., 2021b), VE (Song et al., 2021b), and EDM (Karras et al., 2022), and reduces their training cost by about $50\%$ with comparable image synthesis performance on CIFAR-10, FFHQ, and AFHQv2 datasets. Moreover, FDM decreases their sampling steps by about $3\times$ to achieve similar performance under the same samplers. The codes are in the attached supplementary material and will be released online.

## 1 INTRODUCTION

Diffusion Models (DMs) (Sohl-Dickstein et al., 2015; Ho et al., 2020; Yang et al., 2022) show impressive generative capability in modeling complex data distribution such as the synthesis of image (Dhariwal & Nichol, 2021; Rombach et al., 2022), speech (Kong et al., 2020; Chen et al., 2020), and video (Ho et al., 2022a;b; Khachatryan et al., 2023). However, their slow and costly training and sampling pose significant challenges in broadening their applications. Thus, existing remedies such as loss reweighting (Nichol & Dhariwal, 2021; Hang et al., 2023) and neural network refinement (Song et al., 2021b; Ryu & Ye, 2022) focus on reducing the training cost, while distilled training (Salimans & Ho, 2022; Song et al., 2023) and efficient samplers (Song et al., 2021a; Lu et al., 2022a) target at fewer sampling steps. Despite their efficacy, they are essentially post-hoc modifications and do not delve into the inherent mechanism of DMs: diffusion process, which can accelerate both training and sampling fundamentally.

Our **first contribution** is a novel perspective on the diffusion process of DMs through the lens of stochastic optimization. We find that the forward diffusion process $\mathbf{x}_{t+1} = \alpha_t \mathbf{x}_t + \beta_t \epsilon_t$ ($\epsilon_t \sim \mathcal{N}(0, \mathbf{I})$) coincides with stochastic gradient descent (SGD) (Robbins & Monro, 1951) in optimizing a stochastic time-variant function $f(\mathbf{x}) = \frac{1}{2}\mathbb{E}_{\zeta \sim \mathcal{N}(0, b\mathbf{I})} \|\mathbf{x} - \frac{\beta_t}{1-\alpha_t}\zeta\|_2^2$. At the $t$-th iteration, it samples a minibatch samples $\{\zeta_k\}_{k=1}^b$ to compute stochastic gradient $\mathbf{g}_t = \mathbf{x}_t - \frac{\beta_t}{b(1-\alpha_t)}\sum_{k=1}^b \zeta_k = \mathbf{x}_t - \frac{\beta_t}{1-\alpha_t}\epsilon_t$, and then updates parameter $\mathbf{x}$ via the following stochastic optimization process:

$$\mathbf{x}_{t+1} = \mathbf{x}_t - (1 - \alpha_t)\mathbf{g}_t = \alpha_t\mathbf{x}_t + \beta_t\epsilon_t, \tag{1}$$

where learning rate $(1 - \alpha_t) > 0$ is to align the diffusion and SGD processes. We will detail the connection in Section 4.1.

Our **second contribution** is the development of a novel *Fast Diffusion Model (FDM)*, which accelerates DMs by incorporating momentum SGD (Polyak, 1964) into the diffusion process. At

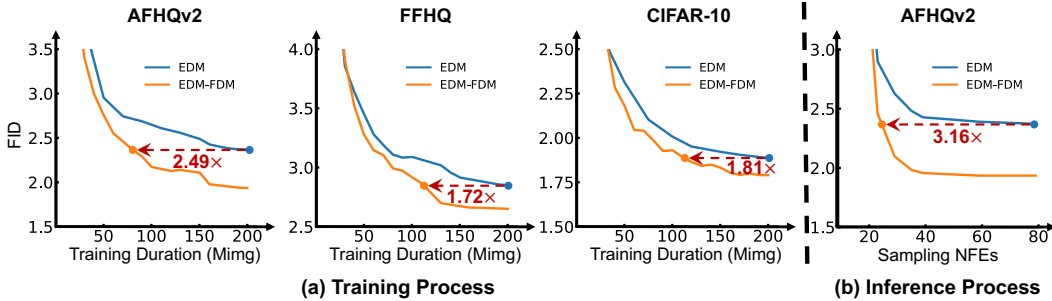

Figure 1: Training and inference processes of EDM and our EDM-FDM. With momentum, EDM-FDM achieves $2\times$ training acceleration on average, and $3.16\times$ sampling acceleration on AFHQv2.

the $t$-th update, momentum SGD not only utilizes a gradient but also an additional momentum $(\mathbf{x}_t - \mathbf{x}_{t-1})$. This momentum accumulates all previous gradients, providing a more stable descent direction compared to the single gradient $\mathbf{g}_t$ used in SGD. Consequently, it effectively reduces solution oscillation, resulting in faster convergence than traditional SGD in both theory and practice (Bollapragada et al., 2022; Loizou & Richtárik, 2017; Sebbouh et al., 2021). Thanks to the equivalence in Eq. (1), in Section 4.2, we are motivated to add the momentum to the forward diffusion process of DMs for faster convergence to the target distribution:

$$\mathbf{x}_{t+1} = \alpha_t \mathbf{x}_t + \beta_t \epsilon_t + \gamma(\mathbf{x}_t - \mathbf{x}_{t-1}), \tag{2}$$

where $\gamma > 0$ is a constant that controls the weight of momentum. Adding the momentum also accelerates the reverse process, *i.e.*, DM sampling, since the reverse process is determined by the forward process and thus gains the same acceleration. Therefore, adding the momentum can accelerate both training and sampling.

However, as we will discuss in Section 4, the crucial perturbation kernel $p(\mathbf{x}_t|\mathbf{x}_0)$ that is required for efficient training and sampling cannot be easily derived by the momentum-based diffusion process in Eq. (2). To this end, we leverage recent advances in DMs to convert this discrete diffusion process to its continuous form (Karras et al., 2022), thereby deriving an analytical solution for the perturbation kernel (Section 4.2). So far, we can plug the momentum diffusion process of Eq. (2) into several popular and effective diffusion models, including VP (Song et al., 2021b), VE (Song et al., 2021b), and EDM (Karras et al., 2022), and build our corresponding FDM (Section 4.3).

Extensive experimental results in Section 5 show that for representative and popular DMs, including EDM, VP and VE, our FDM greatly accelerates their training process by about $2\times$, and improves their sample generation process by about $3\times$. For example, as shown in Figure 1(a), on three datasets, our EDM-FDM uses about half the training cost (million training samples, Mimg) to achieve similar image synthesis performance of EDM. Moreover, for the sample generation process, EDM-FDM achieves similar performance as EDM by using about $1/3$ inference cost in Figure 1(b).

## 2 RELATED WORK

Diffusion-based generative models (DMs) (Sohl-Dickstein et al., 2015; Song & Ermon, 2019; Song et al., 2021b; Ho et al., 2020; Yang et al., 2022) are powerful tools for complex data modeling and generation. Their robust and stable capabilities for complex data modeling have also led to their successful application in various domains, such as text-to-video generation (Ho et al., 2022a;b; Khachatryan et al., 2023), image synthesis (Dhariwal & Nichol, 2021; Ramesh et al., 2022; Gu et al., 2022), manipulation (Rombach et al., 2022; Lugmayr et al., 2022; Song et al., 2023), and audio generation (Kong et al., 2020; Chen et al., 2020; Mittal et al., 2021), *etc*. However, their slow and costly training and sampling limit broader applications. To address these efficiency issues, two family of methods have been proposed. One is known as sampling-efficient DMs, including learning-free sampling and learning-based sampling. Learning-free sampling relies on discretizing the reverse-time SDE (Song et al., 2021b; Dockhorn et al., 2021) or ODE (Lu et al., 2022a;b; Song et al., 2021a; Karras et al., 2022), while learning-based sampling mainly depends on knowledge distillation (Meng et al., 2022; Salimans & Ho, 2022; Song et al., 2023). The other family comprises training-efficient DMs. These methods involve optimization to the loss function (Gu et al., 2022; Karras et al., 2022; Nichol & Dhariwal, 2021; Hang et al., 2023; Song & Ermon, 2020), latent space training (Vahdat

et al., 2021; Gao et al., 2023), neural network refinement (Song et al., 2021b; Ryu & Ye, 2022), and diffusion process improvement (Dockhorn et al., 2021; Das et al., 2023; Lipman et al., 2022). Contrasted with our FDM, other momentum-based approaches in diffusion process improvement necessitate an augmented momentum space (Dockhorn et al., 2021; Pandey & Mandt, 2023), and thus double the memory consumption compared to conventional DMs (Section 5.3). Moreover, these prior works are known to suffer from numerical instability due to the complex formulation of the perturbation kernel. In contrast, like most DMs, our FDM solely possesses the sample space and has a streamlined perturbation kernel, and our FDM enjoys better performance as shown in Section 5.3.

## 3 PRELIMINARIES

**Momentum SGD.** Let us consider an optimization problem, $\min_{\mathbf{x} \in \mathbb{R}^n} f(\mathbf{x})$, where $f$ is a differentiable function. Then, one often uses stochastic gradient descent (SGD) (Robbins & Monro, 1951) to update the parameter $\mathbf{x}$ per iteration:

$$\mathbf{x}_{k+1} = \mathbf{x}_k - \alpha \mathbf{g}_k, \tag{3}$$

where $\mathbf{g}_k$ denotes the stochastic gradient, and $\alpha$ is a step size. Momentum SGD (Polyak, 1964) is proposed to accelerate the convergence of SGD:

$$\mathbf{x}_{k+1} = \mathbf{x}_k - \alpha \mathbf{g}_k + \beta(\mathbf{x}_k - \mathbf{x}_{k-1}) = \mathbf{x}_k - \alpha \mathbf{g}_k - \alpha \sum_{i=0}^{k-1} \beta^{k-i} \mathbf{g}_i, \tag{4}$$

where $\beta$ is a constant. The momentum $\beta(\mathbf{x}_k - \mathbf{x}_{k-1}) = -\sum_{i=0}^{k-1} \alpha \beta^{k-i} \mathbf{g}_i$ accumulates all the past gradients and thus provides a more stable descent direction than the single gradient $\mathbf{g}_t$ used in SGD. Consequently, momentum SGD can effectively avoid oscillation which often exists around the loss regions of high geometric curves, and thus achieves much faster convergence speed than SGD in both theory and practice (Bollapragada et al., 2022; Loizou & Richtárik, 2017; Sebbouh et al., 2021).

**Diffusion Models (DMs).** DMs consist of a forward diffusion process and a corresponding reverse process (Ho et al., 2020). For the forward process, DMs gradually add Gaussian noises into the vanilla sample $\mathbf{x}_0 \sim p_{\text{data}}(\mathbf{x}_0)$, and generates a series of noisy samples $\mathbf{x}_t$:

$$p(\mathbf{x}_t | \mathbf{x}_0) = \mathcal{N}(\mathbf{x}_t; \mu_t \mathbf{x}_0, \mu_t^2 \sigma_t^2 \mathbf{I}), \tag{5}$$

where $\mu_t$ varies along time step $t$. In particular, $p(\mathbf{x}_t | \mathbf{x}_0)$ is also widely known as the *perturbation kernel* applied to the original sample $\mathbf{x}_0$. For VEs (Song et al., 2021b), $\mu_t \equiv 1$; more generally, for others (Ho et al., 2020; Karras et al., 2022; Rombach et al., 2022), $\lim_{t \to \infty} \mu_t = 0$. The noise level $\sigma_t$ increases monotonically with $t$. Accordingly, one can easily obtain the noisy sample $\mathbf{x}_t$ at time step $t$ by $\mathbf{x}_t = \mu_t(\mathbf{x}_0 + \sigma_t \epsilon_t)$, where $\epsilon_t \sim \mathcal{N}(0, \mathbf{I})$ denotes a Gaussian noise. We use $\hat{\mathbf{x}}_t = \mathbf{x}_0 + \sigma_t \epsilon_t$ to denote the non-scaled (w/o $\mu_t$) noisy sample throughout the paper unless specified.

The reverse process from a Gaussian noise (*i.e.* $\mathbf{x}_T$) to the clean sample $\mathbf{x}_0$ is called sampling. Given the score function $\nabla \log p_t(\hat{\mathbf{x}}_t; \sigma_t)$ which indicates the direction of the higher data density (Song et al., 2021b), the reverse process is formulated by a probability flow ODE (Karras et al., 2022):

$$d\mathbf{x} = -\dot{\sigma}_t \sigma_t \nabla \log p_t(\hat{\mathbf{x}}_t; \sigma_t) dt, \tag{6}$$

where $\dot{\sigma}_t$ denotes a time derivative of $\sigma_t$. Given any noise $\hat{\mathbf{x}}_T \sim \mathcal{N}(0, \sigma_T^2 \mathbf{I})$, one can solve Eq. (6) via any numerical ODE solver, and generate real sample $\hat{\mathbf{x}}_0 \sim p_{\text{data}}(\mathbf{x})$. Since the score function $\nabla \log p_t(\hat{\mathbf{x}}_t; \sigma_t)$ is inaccessible in general, one can follow Karras et al. (2022) to use a learnable score network $D_\theta(\hat{\mathbf{x}}_t, \sigma_t)$ to estimate it, formulated as: $\nabla \log p_t(\hat{\mathbf{x}}_t; \sigma_t) = (D_\theta(\hat{\mathbf{x}}_t, \sigma_t) - \hat{\mathbf{x}}_t)/\sigma_t^2$.

For sampling quality (Ho et al., 2020; Song et al., 2021a; Zhang et al., 2022; Rombach et al., 2022), the score network is often reparameterized to predict the noise. For example, in VP (Song et al., 2021b), the score network is defined as $D_\theta(\hat{\mathbf{x}}_t, \sigma_t) := \hat{\mathbf{x}}_t - \sigma_t F_\theta(\mu_t \hat{\mathbf{x}}_t, t)$, where a network $F_\theta$ parameterized by $\theta$ predicts the noise $\epsilon_t$. Then one can train $F_\theta$ via minimizing the loss:

$$\mathcal{L}(D_\theta; t) := \mathbb{E}_{\mathbf{x}_0 \sim p_{\text{data}}(\mathbf{x})} \mathbb{E}_{\mathbf{x}_t \sim p(\mathbf{x}_t | \mathbf{x}_0)} \left[ \lambda(\sigma_t) \| D_\theta(\hat{\mathbf{x}}_t, \sigma_t) - \mathbf{x}_0 \|_2^2 \right], \tag{7}$$

where $\lambda(\sigma_t)$ denotes the standard loss weight to balance the loss at different noise levels $\sigma_t$ (Nichol & Dhariwal, 2021; Karras et al., 2022; Hang et al., 2023).

## 4 METHODOLOGY

Here we first establish a theoretical connection between the forward diffusion process of DDPM with SGD in Section 4.1, and then propose our momentum-based diffusion process in Section 4.2. Finally, we derive the Fast Diffusion Model (FDM) based on the proposed momentum-based diffusion process and elaborate on how to apply FDM to existing diffusion frameworks in Section 4.3.

### 4.1 CONNECTION BETWEEN DDPM AND SGD

Considering a stochastic quadratic time-variant function $f(\mathbf{x}) = \mathbb{E}_{\zeta \sim \mathcal{N}(0,b\mathbf{I})} \frac{1}{2} \|\mathbf{x} - \frac{\beta_t}{1-\alpha_t} \zeta\|_2^2$, where $\beta_t > 0$ and $\alpha_t > 0$ vary with the time step $t$, and $\zeta$ is a standard Gaussian variable, SGD computes the $t$-step stochastic gradient $\mathbf{g}_t$ on a minibatch of training data $\{\zeta_k\}_{k=1}^b$ and update the variable:

$$\mathbf{g}_t = \mathbf{x}_t - \frac{\beta_t}{b(1-\alpha_t)} \sum\nolimits_{k=1}^b \zeta_k = \mathbf{x}_t - \frac{\beta_t}{1-\alpha_t} \epsilon_t, \tag{8}$$

where $\epsilon_t = \frac{1}{b} \sum_{k=1}^b \zeta_k$ and thus also satisfies $\mathcal{N}(0, \mathbf{I})$. If we use the learning rate $\eta = 1 - \alpha_t$, we can align SGD and DDPM by $\mathbf{x}_{t+1} = \alpha_t \mathbf{x}_t + \beta_t \epsilon_t$ as in Eq. (1). By assuming $\alpha_t^2 + \beta_t^2 = 1$, one can observe that the forward diffusion process of DDPM (Ho et al., 2020) and the stochastic optimization process of Eq. (1) share the same formulation. Based on this connection, we can leverage more advanced optimization algorithms to improve the diffusion process of DMs.

### 4.2 MOMENTUM-BASED DIFFUSION PROCESS

**Discrete momentum-based diffusion process.** Inspired by momentum SGD in Section 3, we can accelerate the diffusion process of DDPM by the momentum-based forward diffusion process as in Eq. (2). Then, we provide a theoretical justification for the faster convergence speed of momentum-based forward diffusion process over the conventional diffusion process in DDPM on the constructed function $f(\mathbf{x}) = \frac{1}{2} \mathbb{E}_{\zeta \sim \mathcal{N}(0,b\mathbf{I})} \|\mathbf{x} - \frac{\beta_t}{1-\alpha_t} \zeta\|_2^2$. This is because this function bridges the connection between DDPM and SGD, and can be used as the acceleration feasibility when using momentum SGD to improve the forward diffusion process of DDPM. We summarize our main results in Theorem 1.

**Theorem 1** (Faster convergence of momentum SGD). *Suppose $\frac{\beta_t}{1-\alpha_t} \leq \sigma \ (\forall\, t)$ holds in the function $f(\mathbf{x}) = \mathbb{E}_{\zeta \sim \mathcal{N}(0,b\mathbf{I})} \frac{1}{2} \|\mathbf{x} - \frac{\beta_t}{1-\alpha_t} \zeta\|_2^2$. Define the learning rate as $\eta_t = 1 - \alpha_t$ and denote the initial error as $\Delta = \|\mathbf{x}_0 - \mathbf{x}^*\|_2$, where $\mathbf{x}_0$ is a starting point shared by momentum SGD or vanilla SGD, and $\mathbf{x}^* = 0$ is the optimal mean. Under these conditions, the momentum SGD satisfied $\|\mathbb{E}[\mathbf{x}_k - \mathbf{x}^*]\| \leq \prod_{j=0}^{k-1}(1 - \sqrt{\eta_j})\Delta$, whereas the vanilla SGD satisfied $\|\mathbb{E}[\mathbf{x}_k - \mathbf{x}^*]\| \leq \prod_{j=0}^{k-1}(1 - \eta_j)\Delta$.*

See its proof in Appendix C. Theorem 1 shows that at any arbitrary step $k$, the mean $\mathbb{E}[\mathbf{x}_k]$ of momentum SGD converges faster to the target mean $\mathbf{x}^* = 0$ than vanilla SGD, since $\prod_{j=0}^{k-1}(1-\sqrt{\eta_j})\Delta < \prod_{j=0}^{k-1}(1 - \eta_j)\Delta$ when $\eta_t = 1 - \alpha_t < 1$. Thanks to the above connection between the DDPM's forward diffusion process and SGD (Section 4.1), one can also expect a faster convergence speed of the momentum-based forward diffusion process.

For more clarity, we establish the specific relationship between $\mathbf{x}_t$ and $\mathbf{x}_0$. Here we follow DDPM's notions to streamline our analysis. Specifically, we reformulate the momentum-based diffusion process into the form $\mathbf{x}_{t+1} = \sqrt{1-\beta}\mathbf{x}_t + \sqrt{\beta}\epsilon_t + \gamma(\mathbf{x}_t - \mathbf{x}_{t-1})$ and the vanilla diffusion process as $\mathbf{x}_{t+1} = \sqrt{1-\beta}\mathbf{x}_t + \sqrt{\beta}\epsilon_t$. Denote by $\sigma_{t+1} = \sqrt{1-\beta}\sigma_t + \sqrt{\beta}\epsilon_t + \gamma(\sigma_t - \sigma_{t-1})$ the total accumulated noise at time $t$, with boundary $\sigma_0 = 0$ and $\sigma_1 = \sqrt{\beta}\epsilon_0$. We summarize our result in Theorem 2.

**Theorem 2** (State transition in diffusion process). *For any $\delta \in (0, 4\sqrt{1 - \sqrt{\alpha}})$ with $\alpha = 1 - \beta$, if $\gamma = 2 - 2\sqrt{1 - \sqrt{\alpha}} - \sqrt{\alpha} + \delta$, then*

$$\mathbf{x}_T = \zeta_T \mathbf{x}_0 + \kappa_T \epsilon + \gamma \sum\nolimits_{t=2}^{T-1} \alpha^{\frac{T-t-1}{2}}(\sigma_t - \sigma_{t-1}), \quad \text{(momentum-based diffusion process)}$$

$$\mathbf{x}_T = \sqrt{\alpha^T}\mathbf{x}_0 + \sqrt{1 - \alpha^T}\epsilon, \quad \text{(vanilla diffusion process)}$$

*where $\epsilon \sim \mathcal{N}(0, \mathbf{I})$, $\zeta_T = O(\sqrt{\gamma^T})$, $\kappa_T = \sqrt{1 - \alpha^T + \gamma^2 \alpha^{T-2}(1-\alpha)}$, $\mathbf{x}_0$ is a starting point.*

See its proof in Appendix D. Theorem 2 shows a clean formulation of $\mathbf{x}_t$ at any time step $t$. By comparing the coefficients of the mean ($\zeta_T < \sqrt{\alpha^T}$ ($\forall \alpha \in (0, 1)$)) and variance ($\kappa_T > \sqrt{1 - \alpha^T}$), Theorem 2 also affirms that our momentum-based approach ensures the convergence acceleration of the forward diffusion process towards the equilibrium, *i.e.*, Gaussian distribution.

In this way, we show that momentum can accelerate the forward diffusion process from both optimization and diffusion aspects. Since the reverse process is decided by its forward diffusion process, one can expect the accelerated speed as the forward one, and also enjoy the acceleration effect. Accordingly, momentum can intuitively accelerate not only the training process decided by both forward and reverse processes, but also the sampling process determined by the reverse process, which are empirically testified by our experiments in Section 5.2.

Although Theorem 2 shows the relationship between $\mathbf{x}_t$ and $\mathbf{x}_0$, it does not fully support efficient training and sampling of DMs, since $\mathbf{x}_t$ depends on the computation of the accumulated noise $\sigma_t$ and $\sigma_{t-1}$ and is indeed computationally expensive. To solve the costly training and sampling issue, we leverage recent advances in DMs to convert this discrete diffusion process to its continuous form, and then derive a tractable analytical solution to compute the desired perturbation kernel $p(\mathbf{x}_t|\mathbf{x}_0)$.

**Continuous momentum-based diffusion process.** To compute the perturbation kernel $p(\mathbf{x}_t|\mathbf{x}_0)$, we follow EDM (Karras et al., 2022) which is a SoTA DM, and separately define sample mean and variance. This is because this separation not only directly adopts the idea of the probability flow ODE during DM sampling — matching specific marginal distributions — but also streamlines the analysis. More specifically, following EDM, we remove the stochastic gradient noise in Eq. (2) because $\epsilon_t$ is independent of the $\mathbf{x}_t$ in the $t$-th diffusion step. For the noise, we will handle it in Section 4.3. In this way, we can obtain the deterministic version of Eq. (2) by denoting $\alpha_t := (1 - s\alpha)$:

$$\mathbf{x}_{t+1} = \mathbf{x}_t - s\alpha\mathbf{x}_t + \gamma(\mathbf{x}_t - \mathbf{x}_{t-1}). \tag{9}$$

where $s$ is the step size, $\alpha > 0$ is a scaling parameter, and $\gamma$ is the momentum parameter. By defining $\mathbf{m}_t = (\mathbf{x}_{t+1} - \mathbf{x}_t)/\sqrt{s}$ and $\gamma = 1 - \beta\sqrt{s}$ with $\beta > 0$, Eq. (9) can be rewritten as:

$$\mathbf{m}_{t+1} = (1 - \beta\sqrt{s})\mathbf{m}_t - \sqrt{s}\alpha\mathbf{x}_t; \quad \mathbf{x}_{t+1} = \mathbf{x}_t + \sqrt{s}\mathbf{m}_t. \tag{10}$$

Let $s \to 0$ in Eq. (10), we obtain the following ODE by inverting the Euler Method (Atkinson, 1991):

$$\frac{\mathrm{d}\mathbf{x}(t)}{\mathrm{d}t} = \mathbf{m}(t); \quad \frac{\mathrm{d}\mathbf{m}(t)}{\mathrm{d}t} = -\beta\mathbf{m}(t) - \alpha\mathbf{x}(t), \tag{11}$$

which corresponds to a second-order ODE of $\ddot{\mathbf{x}}(t) + \beta\dot{\mathbf{x}}(t) + \alpha\mathbf{x}(t) = 0$. Furthermore, we observe that the above ODE is a Damped Oscillation ODE which describes an oscillation system (McCall, 2010). We then follow (Dockhorn et al., 2021) and carefully calibrate the hyper-parameters $\alpha$ and $\beta$ to achieve Critical Damping which allows an oscillation system to return to equilibrium faster without oscillation or overshoot. See the mechanism behind the Critical Damping in (McCall, 2010). In addition, we discuss the overshoot issue that exists in DM and illustrate how this Critical Damping state alleviates it in Appendix B.3. Accordingly, we can obtain a Critically Damped ODE:

$$\ddot{\mathbf{x}}(t) + 2\beta\dot{\mathbf{x}}(t) + \beta^2\mathbf{x}(t) = 0. \tag{12}$$

## 4.3 FAST DIFFUSION MODEL

**Forward diffusion process.** By solving the Critically Damped ODE in Eq. (12) with the boundary conditions $\mathbf{x}(0) = \mathbf{x}_0$ and $\dot{\mathbf{x}}(0) = 0$, we obtain the mean-varying process of $\mathbf{x}$:

$$\mathbf{x}(t) = e^{-\mathcal{B}(t)}(1 + \mathcal{B}(t))\mathbf{x}_0, \tag{13}$$

where $\mathcal{B}(t) = \int_0^t \beta(s)\mathrm{d}s$. Here we follow the works (Ho et al., 2020; Dockhorn et al., 2021) and define $\beta(s)$ as a monotonically increasing linear function, since an iteration-adaptive step size $\beta(s)$ often yields fast convergence while preserving image details at the early stage.

Then we first compute the perturbation kernel $p_{\text{FDM}}(\mathbf{x}_t|\mathbf{x}_0) = \mathcal{N}(\mathbf{x}_t; e^{-\mathcal{B}(t)}(1 + \mathcal{B}(t))\mathbf{x}_0, 0)$ of the deterministic flow in Eq. (13) at time step $t$. Next, we follow EDM (Karras et al., 2022), and incorporate an independent Gaussian noise of variance $\sigma_t^2$ into the perturbation kernel:

$$p_{\text{FDM}}(\mathbf{x}_t|\mathbf{x}_0) = \mathcal{N}(\mathbf{x}_t; e^{-\mathcal{B}(t)}(1 + \mathcal{B}(t))\mathbf{x}_0, \sigma_t^2\mathbf{I}). \tag{14}$$

Table 1: Comparison of score network $D(\mathbf{x}; \sigma_t)$ among VP, VE, EDM, and their FDM versions. The highlighted parts denote the modifications when integrating our FDM into vanilla DMs.

| | $D_\theta(\mathbf{x}; \sigma_t)$ | $D_{\text{FDM}}(\mathbf{x}; \sigma_t)$ (ours) |
|---|---|---|
| **VP** (Song et al., 2021b) | $\mathbf{x} - \sigma_t F_\theta\left(e^{-\frac{1}{2}\int_0^t \beta(s)\mathrm{d}s}\mathbf{x}, t\right)$ | $\mathbf{x} - \sigma_t F_\theta\left( e^{-\int_0^{t'}\beta(s)\mathrm{d}s}(1+\int_0^{t'}\beta(s)\mathrm{d}s)\,\mathbf{x}, t \right)$ |
| **VE** (Song et al., 2021b) | $\mathbf{x} + \sigma_t F_\theta\left(\mathbf{x}, \sigma_t\right)$ | $\mathbf{x} + \sigma_t F_\theta\left( e^{-\int_0^{t'}\beta(s)\mathrm{d}s}(1+\int_0^{t'}\beta(s)\mathrm{d}s)\,\mathbf{x}, \sigma_t \right)$ |
| **EDM** (Karras et al., 2022) | $\frac{\sigma_{\text{data}}^2}{\sigma_t^2+\sigma_{\text{data}}^2}\mathbf{x} + \frac{\sigma_t\sigma_{\text{data}}}{\sqrt{\sigma_t^2+\sigma_{\text{data}}^2}}F_\theta\left(\frac{1}{\sqrt{\sigma_t^2+\sigma_{\text{data}}^2}}\mathbf{x}, \sigma_t\right)$ | $\frac{\sigma_{\text{data}}^2}{\sigma_t^2+\sigma_{\text{data}}^2}\mathbf{x} + \frac{\sigma_t\sigma_{\text{data}}}{\sqrt{\sigma_t^2+\sigma_{\text{data}}^2}}F_\theta\left( e^{-\int_0^{t'}\beta(s)\mathrm{d}s}(1+\int_0^{t'}\beta(s)\mathrm{d}s)\,\mathbf{x}, \sigma_t \right)$ |

In practice, when applying our momentum to a certain DM, we employ the noise schedule $\sigma_t$ in the corresponding DM. This can be interpreted as incorporating the expectation of momentum into the vanilla diffusion process of a certain DM, which facilitates seamless integration of our momentum-based diffusion process with existing DMs. Accordingly, given a sample $\mathbf{x}_0$, one can directly sample its noisy version $\mathbf{x}_t$ at any time step $t$ via Eq. (14) for the forward process. For more stability, we adopt the reparameterize techniques in (Ho et al., 2020), and sample the noisy sample at time step $t$ as

$$\mathbf{x}_t = e^{-\mathcal{B}(t)}(1 + \mathcal{B}(t))\mathbf{x}_0 + \sigma_t \epsilon_t, \tag{15}$$

where $\mathbf{x}_0 \sim p_{\text{data}}(\mathbf{x})$ denotes the real sample, and $\epsilon_t \sim \mathcal{N}(0, \mathbf{I})$ denotes a random Gaussian noise.

**Reverse sampling process.** With the perturbation kernel defined in Eq. (14), we can define our score network $D_{\text{FDM}}(\mathbf{x}, \sigma_t)$, and use it to denoise a noisy sample. A diffusion model, primarily determined by its perturbation kernel $p(\mathbf{x}_t|\mathbf{x}_0)$, can incorporate our momentum-based diffusion process simply by replacing their perturbation kernel with our $p_{\text{FDM}}(\mathbf{x}_t|\mathbf{x}_0)$.

We demonstrate this with popular and effective DMs including VP (Song et al., 2021b), VE (Song et al., 2021b), and EDM (Karras et al., 2022), showcasing the integration of our FDM into these models and building the corresponding score network $D_{\text{FDM}}(\mathbf{x}, \sigma_t)$. As shown in Table 1, the score network $D_\theta(\mathbf{x}, \sigma_t)$ improved by modifying the input parts related to the perturbation kernel. This simple modification already accomplishes the replacement of the vanilla diffusion process with our momentum-based counterpart.

Moreover, in FDM, our time step $t$ is defined within the range $[0, 1]$, contrasting to several models like VE and EDM that use noise level $\sigma$ as the time variable and often exceed our definition range. To this end, we devise a reverse scaling function $\sigma^{-1}(\cdot)$ to project the noise level $\sigma$ into $[0, 1]$ as follows:

$$t' = \sigma^{-1}(\sigma) = \frac{\sigma - \sigma_{\min}}{\sigma_{\max} - \sigma_{\min}}, \tag{16}$$

where the projected variable $t'$ is used as the time step of FDM in Table 1. We adopt this simple linear function due to its consistency with the accumulation law of stochastic gradient noise in SGD as shown in Eq. (8). After defining the score network $D_{\text{FDM}}(\mathbf{x}, \sigma_t)$, we train network $F_\theta$ by minimizing

$$\mathcal{L}(D_{\text{FDM}}; t) := \mathbb{E}_{\mathbf{x}_0 \sim p_{\text{data}}(\mathbf{x})}\mathbb{E}_{\mathbf{x}_t \sim p_{\text{FDM}}(\mathbf{x}_t|\mathbf{x}_0)}\left[\hat{\lambda}(\sigma_t)\|D_{\text{FDM}}(\hat{\mathbf{x}}_t, \sigma_t) - \mathbf{x}_0\|_2^2\right], \tag{17}$$

where $\hat{\lambda}(\sigma) := \min\{\lambda(\sigma), \lambda_{\max} \cdot \tau_k\}$ is to balance the training process. Here $\lambda(\sigma)$ is the standard weight used by vanilla DM in Eq. (7), and $\tau_k := \tau^k$ increases along with training iteration number $k$, where the constant $\tau$ slightly bigger than 1 is to control the increasing speed. This is because as shown in Theorem 2, at time step $t$, the noisy sample $\mathbf{x}_t$ in the momentum-based forward process often contains more noise than the vanilla forward process. Then, for momentum-based diffusion, predicting vanilla sample $\mathbf{x}_0$ from $\mathbf{x}_t$ is more challenging, especially for the early training phase, which may yield abnormally large losses. So we use the clamp weight $\hat{\lambda}(\sigma)$ to reduce the side effects of these abnormal losses, and gradually increase $\hat{\lambda}(\sigma)$ along training iterations where the network progressively becomes stable and better.

With the well-trained score network, $D_{\text{FDM}}(\mathbf{x}, \sigma_t)$, we can estimate the score function $\nabla \log p_t(\mathbf{x}; \sigma_t)$ to generate realistic samples. Here $\nabla \log p_t(\mathbf{x}; \sigma_t)$ already implicitly incorporates our momentum term as evidenced by definition of $p_t(\mathbf{x}; \sigma_t)$ in Table 1, and thus helps faster generation. Indeed, this implicit incorporation allows our FDM to be used into different solvers (Liu et al., 2021; Lu et al., 2022a) as testified in Section 5.2. Following Karras et al. (2022), we set $\sigma_t := t$ during the reverse process and discretize the time horizon $[t_{\min}, t_{\max}]$ into $N - 1$ sub-intervals $t_i =$

Table 2: Image synthesis performance (FID) under different million training images (Mimg). We use official NFEs (number of function evaluations) to synthesize, *e.g.* 35 for CIFAR-10 and 79 for others.

| Dataset | Duration (Mimg) | Method | | | | | |
|---|---|---|---|---|---|---|---|
| | | EDM | EDM-FDM | VP | VP-FDM | VE | VE-FDM |
| CIFAR-10 $32 \times 32$ | 50 | 5.76 | 2.17 | 2.74 | 2.74 | 49.47 | 10.01 |
| | 100 | 1.99 | 1.93 | 2.24 | 2.24 | 4.05 | 3.26 |
| | 150 | 1.92 | 1.83 | 2.19 | 2.13 | 3.27 | 3.00 |
| | 200 | 1.88 | 1.79 | 2.15 | 2.08 | 3.09 | 2.85 |
| FFHQ $64 \times 64$ | 50 | 3.21 | 3.27 | 3.07 | 12.49 | 96.49 | 93.72 |
| | 100 | 2.87 | 2.69 | 2.83 | 2.80 | 94.14 | 88.42 |
| | 150 | 2.69 | 2.63 | 2.73 | 2.53 | 79.20 | 4.73 |
| | 200 | 2.65 | 2.59 | 2.69 | 2.43 | 38.97 | 3.04 |
| AFHQv2 $64 \times 64$ | 50 | 2.62 | 2.73 | 3.46 | 25.70 | 57.93 | 54.41 |
| | 100 | 2.57 | 2.05 | 2.81 | 2.65 | 57.87 | 52.45 |
| | 150 | 2.44 | 1.96 | 2.72 | 2.47 | 57.69 | 50.53 |
| | 200 | 2.37 | 1.93 | 2.61 | 2.39 | 57.48 | 47.30 |

$(t_{\min}^{\frac{1}{\rho}} + \frac{i}{N-1}(t_{\max}^{\frac{1}{\rho}} - t_{\min}^{\frac{1}{\rho}}))^{\rho}$ with $\rho = 7$. Accordingly, we denoise the noisy sample $\hat{\mathbf{x}}_{t_{i+1}}$ to obtain a more clean sample $\hat{\mathbf{x}}_{t_i}$ via the following discrete reverse sampling process of FDM:

$$\hat{\mathbf{x}}_{t_i} = \hat{\mathbf{x}}_{t_{i+1}} + (t_i - t_{i+1})(\hat{\mathbf{x}}_{t_{i+1}} - D_{\text{FDM}}(\hat{\mathbf{x}}_{t_{i+1}}; t_{i+1}))/t_{i+1}. \qquad (18)$$

By iteratively computing the sample starting from $\hat{\mathbf{x}}_T \sim \mathcal{N}(0, \sigma_T^2 \mathbf{I})$ in a reverse temporal order as Eq. (18), we can eventually estimate a sample $\hat{\mathbf{x}}_0$ which is often realistic sample drawn from $p_{\text{data}}(\mathbf{x})$.

## 5 EXPERIMENTS

**Implementation Details.** We integrate our momentum-based diffusion process into various models including VP (Song et al., 2021b), VE (Song et al., 2021b), and EDM (Karras et al., 2022), and build our corresponding FDMs, *i.e.*, VP-FDM, VE-FDM, and EDM-FDM respectively. For VP and VE, we follow their vanilla network architectures (Song et al., 2021b), DDPM++ and NCSN++, across all datasets. For EDM, we utilize their officially modified DDPM++. In our FDM, we retain these official architectures, and hyper-parameter setting from EDM, and we also adopt their default Adam (Kingma & Ba, 2014) optimizer to train each model by a total of 200 million images (Mimg). We initially set $\lambda_{\max} = 5$ and compute $\tau$ according to the training batch size to ensure that $\lambda_{\max} \cdot \tau^k$ reaches 500 after the model has been trained with $10,000$ images. See more details in Appendix B.

**Evaluation Setting.** We test the popular scenarios, including unconditional image generation on **AFHQv2** (Choi et al., 2020) and **FFHQ** (Karras et al., 2019), and the conditional image generation on **CIFAR-10** (Krizhevsky, 2009). We follow EDM to use $64 \times 64$-sized **AFHQv2** and **FFHQ**. For evaluation, we use Exponential Moving Average (EMA) models to generate $50,000$ images using the EDM sampler based on Heun's $2^{\text{nd}}$ order method (Atkinson, 1991), and report FID score.

### 5.1 TRAINING COST COMPARISON

Table 2 shows that on the three datasets, FDMs consistently improve the vanilla diffusion models, *i.e.* VP, VE, and EDM, under the same training samples and the same sampling NFEs. Specifically, EDM-FDM makes $0.20$ average FID improvement over EDM on the three datasets. Similarly, VE-FDM and VP-FDM also improves their corresponding VE and VP by $15.45$ and $0.18$ average FID, respectively. The big improvement on VE is because the vanilla VE is not stable and often fails to achieve good performance as reported in EDM (Karras et al., 2022), while our momentum-based diffusion process uses momentum term which indeed accumulates the past historical information, *e.g.* noise injection and sampling-decaying, and thus provides more stable synthesis guidance. As a result, VE-FDM greatly improves VE.

More importantly, FDMs also consistently boost the convergence or learning speed of the vanilla VP, VE, and EDM. For instance, to achieve the same or lower FID score of EDM using 200M training samples, EDM-FDM only needs about 110M, 110M, and 80M training images on CIFAR10, FFHQ, and AFHQv2, respectively, yielding about $2\times$ faster learning speed. As shown by Figure 2, on VP and VE frameworks, one can also observe the learning acceleration of our FDM.

Table 3: Image synthesis performance (FID) under different inference costs (number of function evaluations, NFEs) on AFHQv2 with EDM sampler. All models are trained on 200 Mimg.

| Method NFE | EDM | EDM-FDM | VP | VP-FDM | VE | VE-FDM |
|---|---|---|---|---|---|---|
| 25 | 2.78 | 2.32 | 2.88 | 2.59 | 61.04 | 48.29 |
| 49 | 2.39 | 1.93 | 2.64 | 2.41 | 57.59 | 47.49 |
| 79 | 2.37 | 1.93 | 2.61 | 2.39 | 57.48 | 47.30 |

Table 4: Image synthesis performance (FID) under different inference costs (number of function evaluations, NFEs) on AFHQv2 with DPM-Solver++. All models are trained on 200 Mimg.

| Method NFE | EDM | EDM-FDM | VP | VP-FDM | VE | VE-FDM |
|---|---|---|---|---|---|---|
| 25 | 2.60 | 2.09 | 2.99 | 2.64 | 59.26 | 49.51 |
| 49 | 2.42 | 1.98 | 2.79 | 2.45 | 59.16 | 48.68 |
| 79 | 2.39 | 1.95 | 2.78 | 2.42 | 58.91 | 48.66 |

Indeed, VP-FDM and VE-FDM on average improve the learning speed of VP and VE by $1.66\times$ and $2.06\times$ respectively across the three datasets. All these results demonstrate the superiority of our FDM in terms of learning acceleration and compatibility with different diffusion models.

We also observe that VP-FDM performs worse than VP when using 50M training samples in Table 2. This is because, at the beginning of training, the momentum term in VP-FDM does not accumulate sufficient historical information, and is not very stable, especially on the complex AFHQv2 dataset which contains diverse animal faces. But once VP-FDM sees enough training samples, its momentum becomes stable and its performance is improved quickly, surpassing vanilla VP using 200M training samples by using only almost half training samples. This is indeed observed in Figure 3.

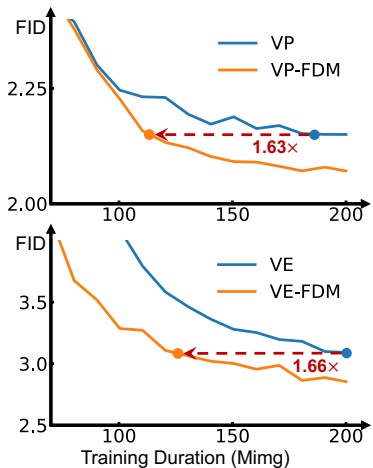

Figure 2: Training acceleration of FDM on VP & VE on CIFAR-10.

## 5.2 INFERENCE COST COMPARISON

Here we compare the performance under the different number of function evaluations (NFEs, a.k.a sampling steps). For fairness, all models are trained on 200 million images (Mimg). Both the FDM (*e.g.*, EDM-FDM) and the baselines (*e.g.*, EDM) share the same state-of-the-art (SoTA) EDM sampler for alignment. The results are listed in Table 3. Moreover, our results are consolidated by evaluations based on another SoTA numerical solver, DPM-Solver++ (Lu et al., 2022b) in Table 4.

From our experimental results, it is evident that under the same NFEs, our FDM consistently outperforms baseline DMs, including VP, VE, and the SoTA EDM. Notably, EDM-FDM, with only 25 NFEs, surpasses the performance of EDM, which requires 79 NFEs, translating to an acceleration of $3.16\times$. This efficiency improvement is mirrored in both VP-FDM and VE-FDM. Furthermore, when evaluating our FDM using the DPM-Solver++, the results suggest that our FDM is a robust and efficient framework that capable of improving the image sampling process across a wide range of diffusion models and advanced numerical solvers.

## 5.3 ABLATION STUDY

**Loss weight warm-up strategy.** Our loss weight warm-up strategy is designed to stabilize FDM training during the initial training stage. This is because, at the early training phase, the network model is not stable in the vanilla DMs, and would become worse when plugging momentum into the diffusion process, since momentum indeed yields more noisy $\mathbf{x}_t$ than vanilla DMs, and enforcing the model to predict the bigger noise in $\mathbf{x}_t$ leads to unstable training caused by possible abnormal training loss, *etc*. To address this issue, we propose the loss weight warm-up strategy to gradually increase the weight of training loss along with training iterations. Table 5 shows an improvement given by our loss weight warm-up strategy during the early stages of training on CIFAR-10.

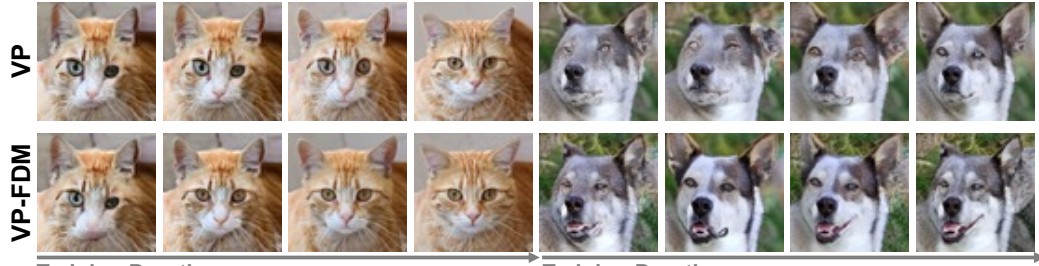

Figure 3: Qualitative comparison of the synthesized images by VP and VP-FDM on AFHQv2 dataset. Images in each column are synthesized when models are respectively trained on 50, 100, 150, and 200 million images (Mimg). By comparison, our FDM significantly accelerates the training process.

Table 5: Effects of our proposed loss weight.

| Method | Duration (Mimg) | | | |
|---|---|---|---|---|
| | 50 | 100 | 150 | 200 |
| EDM | 5.76 | 1.99 | 1.92 | 1.88 |
| EDM w/ $\hat{\lambda}(\sigma)$ | 2.14 | 1.92 | 1.89 | 1.87 |
| FDM w/o $\hat{\lambda}(\sigma)$ | 2.29 | 1.95 | 1.85 | 1.82 |
| FDM | 2.17 | 1.93 | 1.83 | 1.79 |

Table 6: Ablation study on step size.

| Method | Duration (Mimg) | | | |
|---|---|---|---|---|
| | 50 | 100 | 150 | 200 |
| VP | 2.74 | 2.24 | 2.19 | 2.15 |
| VP 2× | 3.12 | 2.49 | 2.36 | 2.29 |
| VP 3× | 3.51 | 2.80 | 2.54 | 2.42 |
| VP-FDM | 2.74 | 2.24 | 2.13 | 2.08 |

**Momentum v.s. faster converting sample into Gaussian noise via larger step size.** To fast converse one sample into a Gaussian noise in the forward diffusion process, we test VP on CIFAR-10 by using double or triple step size function $\beta(t)$ in Table 1, respectively denoted by "VP 2×" and "VP 3×". Table 6 shows that an increasing step size fails to accelerate model training and yields inferior results. In contrast, our FDM accelerates the training process by accumulating historical information, and adaptively adjusting the step size during different diffusion stages. Along with the forward diffusion process, the momentum in FDM accumulates more noise and gradually becomes large, while in the later diffusion process, momentum gradually decreases since the image is approaching the Gaussian noise. So this adaptive step size in FDM according to the diffusion process is superior to the approach of simply increasing step size to fast convert a sample into a Gaussian noise.

**Comparing to other momentum-based approach.** We compared CLD (Dockhorn et al., 2021) with our FDM on CIFAR-10 in Table 7. Here VP-FDM and CLD employ identical network architectures (i.e., DDPM++), time distribution (i.e., uniformly distributed $t$), formulation of step size $\beta_t$, and comparable training iterations. The experimental results show that our VP-FDM achieves superior performance than CLD. Moreover, compared to our

Table 7: Comparison between FDM and CLD.

| Method | Params | Sampler | NFE | FID |
|---|---|---|---|---|
| CLD | 1076M | RK45 | 302 | 2.29 |
| VP-FDM | 557M | RK45 | 113 | 2.03 |
| CLD | 1076M | EDM | 300 | 26.81 |
| CLD | 1076M | EDM | 35 | 82.35 |
| VP-FDM | 557M | EDM | 35 | 2.08 |

approach, CLD requires double the computational resources since it processes both data **x** and velocity **v**, posing challenges to integrate it with SoTA DMs which often have large models.

## 6 CONCLUSION

In this work, we accelerate the diffusion process of diffusion models through the lens of stochastic optimization. We first establish the connections between the diffusion process and the stochastic optimization process of SGD. Then considering the faster convergence of momentum SGD over SGD, we use the momentum in momentum SGD to accelerate the diffusion process of diffusion models, and also show the theoretical acceleration. Moreover, we integrate our improved diffusion processes with several diffusion models for acceleration, and derive their score functions for practical usage. Experimental results show our fast diffusion model improves VP (Song et al., 2021b), VE (Song et al., 2021b), and EDM (Karras et al., 2022) by accelerating their learning speed by at least 1.6× and also achieving much faster inference speed via largely reducing the sampling steps.

## REPRODUCIBILITY STATEMENT

To ensure the rigor and reproducibility of our work, we provide access to all essential resources necessary for accurate replication of our experiments:

- **Code Access**: Our comprehensive codebase is provided in the supplementary ZIP file, and we intend to share it with the wider research community in the near future.
- **Model Checkpoints**: Along with the code, we commit to releasing the associated model checkpoints to enable exact replication and further research.
- **Datasets**: All datasets we used are publicly available, which ensures easy replication of our experiments.

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

# Appendices

## A  LIMITATION

Firstly, while the FDM is designed for use with general DMs, our verification is limited to three popular DMs. Secondly, here we only evaluate our method on several datasets which may not well explore the performance of our method, as we believe our FDM could play a significant role in reducing both training and sampling costs in various tasks.

## B  EXPERIMENTAL RESULTS

### B.1  ADDITIONAL IMPLEMENTATION DETAILS

We implemented our Fast Diffusion Model and the corresponding baselines based on EDM (Karras et al., 2022) codebase. We follow the default hyper-parameter settings for a fair comparison across all models. Specifically, for CIFAR-10, we use Adam optimizer with the learning rate $1 \times 10^{-3}$ and batch size 512; for FFHQ and AFHQv2, we use learning rate $2 \times 10^{-4}$ and batch size 256.

Across all datasets, we adopt a learning rate ramp-up duration of 10 Mimgs, and we set the EMA half-life as 0.5 Mimgs. We initially set $\lambda_{\max} = 5$ and compute $\tau$ according to the training batch size to ensure that $\lambda_{\max} \cdot \tau^k$ reaches 500 after the model has been trained with $10,000$ images. Thus, we set $\tau = 1.023$ on CIFAR-10 and $\tau = 1.011$ on other datasets based on different batch size.

We ran all experiments using PyTorch 1.13.0, CUDA 11.7.1, and CuDNN 8.5.0 with 8 NVIDIA A100 GPUs.

### B.2  ADDITIONAL RESULTS

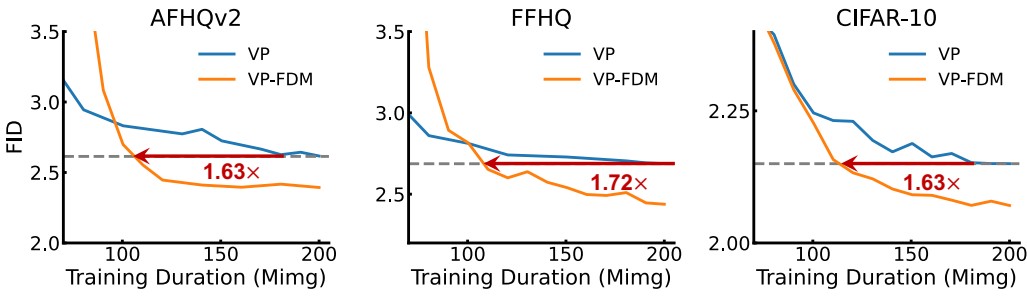

Figure 4: Training processes of VP and our VP-FDM. With momentum, VP-FDM achieves $1.66\times$ training acceleration on average.

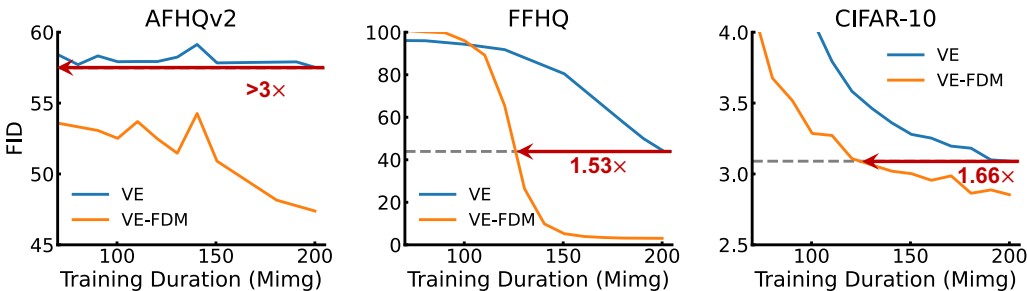

Figure 5: Training processes of VE and our VE-FDM. With momentum, VE-FDM achieves $2.06\times$ training acceleration on average.

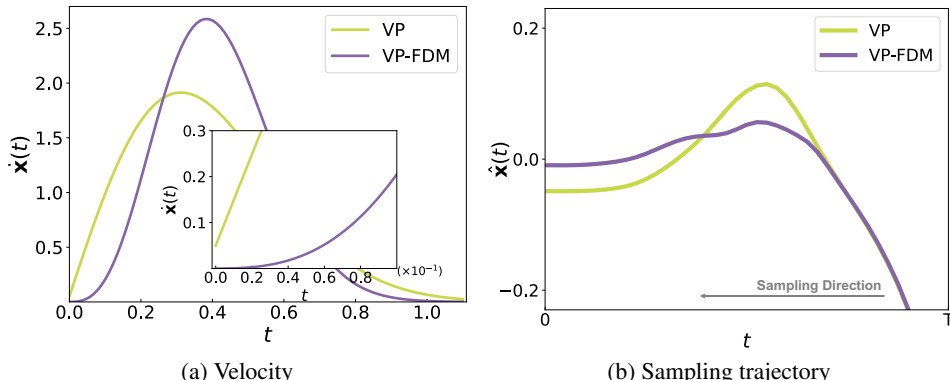

(a) Velocity            (b) Sampling trajectory

Figure 6: Comparison between VP and our VP-FDM. **(a)** The observed velocity of VP around $t = 0$ fluctuates rapidly and fails to converge to 0. **(b)** The trajectory of probability ODE of VP and VP-FDM at random pixel locations. The non-converging velocity of VP leads to an overshoot issue during sampling, while our VP-FDM successfully mitigates this issue.

Table 8: Training time (hours) under different million training images (Mimg) on AFHQv2 dataset.

| Duration (Mimg) | EDM | EDM-FDM | VP | VP-FDM | VE | VE-FDM |
|---|---|---|---|---|---|---|
| 50 | 19.2 | 19.3 | 19.3 | 19.4 | 19.5 | 19.6 |
| 100 | 38.6 | 38.9 | 38.9 | 39.0 | 39.5 | 39.5 |
| 150 | 58.1 | 58.3 | 58.5 | 58.6 | 59.3 | 59.3 |
| 200 | 77.3 | 77.6 | 78.0 | 78.0 | 78.9 | 78.8 |

(a) EDM    FID:1.88                (b) EDM-FDM    FID:1.79

(c) VP    FID:2.15                (d) VP-FDM    FID:2.08

(e) VE    FID:3.09                (f) VE-FDM    FID:2.85

Figure 7: Results for different diffusion models of the same set of initial points $(\mathbf{x}_T)$ on conditional CIFAR-10 with 35 NFEs.

We further show more qualitative results of the models in the main paper. we provide the training process of VP and VE in Figure 4 and Figure 5. Moreover, we compare the conditional CIFAR-10 in Figure 7, unconditional FFHQ and AFHQv2 in Figure 8 and Figure 9, respectively. These illustrations consistently justify our discussions in the main paper.

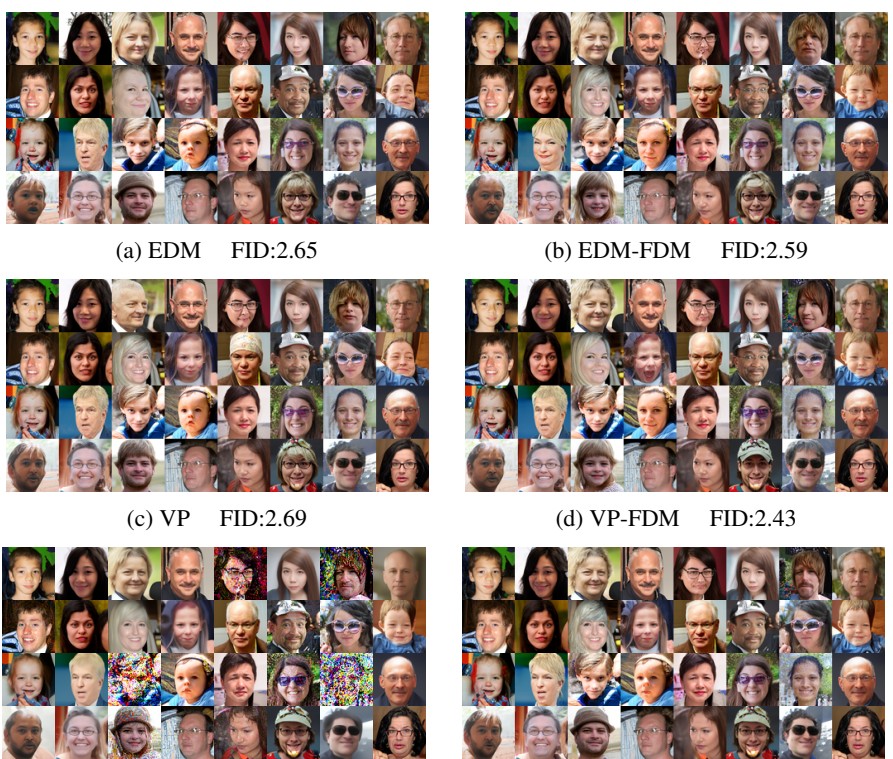

(a) EDM    FID:2.65

(b) EDM-FDM    FID:2.59

(c) VP    FID:2.69

(d) VP-FDM    FID:2.43

(e) VE    FID:38.97

(f) VE-FDM    FID:3.04

Figure 8: Results for different diffusion models of the same set of initial points ($\mathbf{x}_T$) on FFHQ with 79 NFEs.

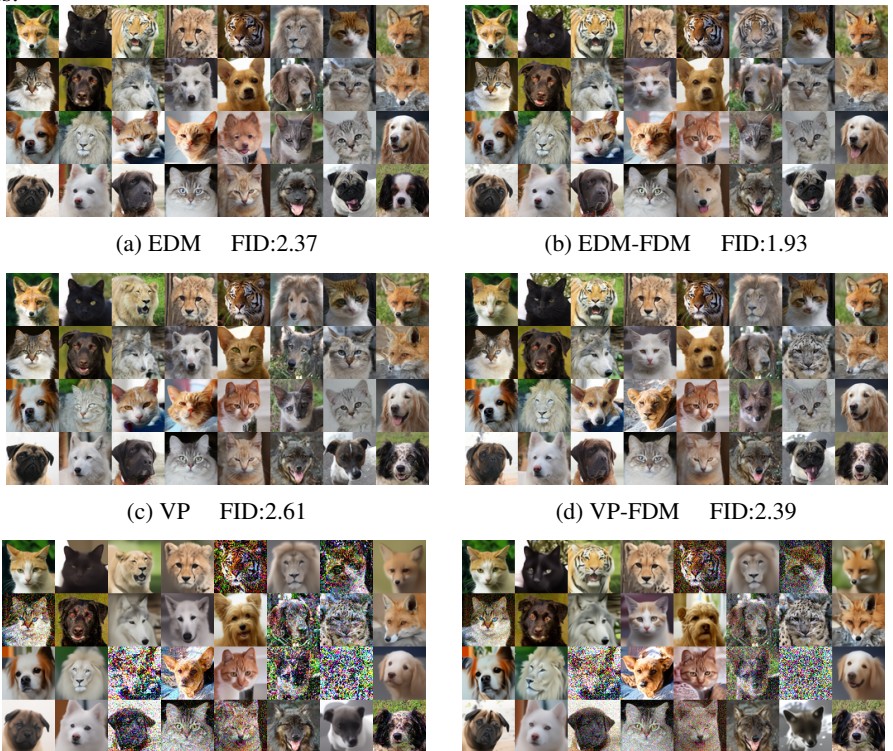

(a) EDM    FID:2.37

(b) EDM-FDM    FID:1.93

(c) VP    FID:2.61

(d) VP-FDM    FID:2.39

(e) VE    FID:57.48

(f) VE-FDM    FID:47.30

Figure 9: Results for different diffusion models of the same set of initial points ($\mathbf{x}_T$) on AFHQv2 with 79 NFEs.

We also conducted an analysis of the training time of our FDM, compared to other baseline DMs as shown in Table 8. The experimental results indicate that our method does not increase the training time per iteration.

### B.3 FDM BENEFITS THE SAMPLING PROCESS

Additionally, we discuss the benefits of our FDM in the sampling process by taking the classical and effective VP (Song et al., 2021b) as an example. Specifically, the corresponding velocity $\dot{\mathbf{x}}(t)$ of VP and FDM can be written as

$$\dot{\mathbf{x}}_{\text{VP}}(t) = -\frac{1}{2}\beta(t)\exp(-\frac{1}{2}\mathcal{B}(t))\mathbf{x}_0; \quad \dot{\mathbf{x}}_{\text{FDM}}(t) = -\beta(t)\mathcal{B}(t)\exp(-\mathcal{B}(t))\mathbf{x}_0. \tag{19}$$

By comparison, one can observe that the boundary velocity in FDM approaches zero at both $t \to 0$ and $t \to \infty$, a property not shared by the VP model as depicted in Figure 6a. The velocity of VP around 0 not only fluctuates rapidly but also fails to converge to zero, thereby negatively affecting the image generation process. Specifically, as illustrated in Figure 6b, the image generation process of VP suffers from an overshoot issue where the predicted pixels surpass their target value before returning to the desired value. In contrast, our FDM effectively mitigates this issue. By applying an additional boundary condition on velocity, we ensure convergence. Moreover, the smooth fluctuation of momentum around zero provides a more stable image generation process, which potentially allows for an increase in the sampling step size.

## C PROOF OF THEOREM 1

**Theorem 1.** *Suppose $\frac{\beta_t}{1-\alpha_t} \le \sigma$ ($\forall$ $t$) holds in the function $f(\mathbf{x}) = \mathbb{E}_{\zeta \sim \mathcal{N}(0, b\mathbf{I})}\frac{1}{2}\|\mathbf{x} - \frac{\beta_t}{1-\alpha_t}\zeta\|_2^2$. Define the learning rate as $\eta_t = 1 - \alpha_t$ and denote the initial error as $\Delta = \|\mathbf{x}_0 - \mathbf{x}^*\|_2$, where $\mathbf{x}_0$ is a starting point shared by momentum SGD or vanilla SGD, and $\mathbf{x}^* = 0$ is the optimal mean. Under these conditions, the momentum SGD satisfied $\|\mathbb{E}[\mathbf{x}_k - \mathbf{x}^*]\| \le \prod_{j=0}^{k-1}(1 - \sqrt{\eta_j})\Delta$, whereas the vanilla SGD satisfied $\|\mathbb{E}[\mathbf{x}_k - \mathbf{x}^*]\| \le \prod_{j=0}^{k-1}(1 - \eta_j)\Delta$.*

*Proof.* **Vanilla SGD:** Consider $\mathbf{x} \in \mathbb{R}[-1, 1]^d$. Let $\sigma_t = \frac{\beta_t}{1-\alpha_t} = \sqrt{\frac{2-\eta_t}{\eta_t}}$.

For vanilla SGD, we have:
$$\mathbf{x}_{k+1} = \mathbf{x}_k - \eta_k(\mathbf{x}_k - \sigma_k\epsilon_k), \tag{20}$$
where $\epsilon_k = \frac{1}{b}\sum_{i=1}^b \zeta_i$ denotes the mean of a sampled minibatch $\{\zeta_i\}_{i=1}^b$, thus satisfied $\epsilon_k \sim \mathcal{N}(0, \mathbf{I})$. Hence, for the optimum mean $\mathbf{x}^* = 0$, we have

$$\mathbf{x}_{k+1} - \mathbf{x}^* = \mathbf{x}_k - \mathbf{x}^* - \eta_k(\mathbf{x}_k - \mathbf{x}^*) + \eta_k\sigma\epsilon_k \tag{21}$$

$$= \prod_{j=0}^k(1 - \eta_j)(\mathbf{x}_0 - \mathbf{x}^*) + \sum_{j=0}^k\prod_{i=1}^{k-j}(1 - \eta_i)\eta_j\sigma_j\epsilon_j. \tag{22}$$

Denote the initial error as $\Delta = \|\mathbf{x}_0 - \mathbf{x}^*\|$, we derive the norm of expectation error:

$$\|\mathbb{E}[\mathbf{x}_{k+1} - \mathbf{x}^*]\| \le \left\|\prod_{j=0}^k(1 - \eta_j)(\mathbf{x}_0 - \mathbf{x}^*) + \mathbb{E}\left[\sum_{j=0}^k\prod_{i=1}^{k-j}(1 - \eta_i)\eta_j\sigma_j\epsilon_j\right]\right\| \tag{23}$$

$$\le \prod_{j=0}^k(1 - \eta_j)\|\mathbf{x}_0 - \mathbf{x}^*\| + \sigma\sum_{j=0}^k\prod_{i=1}^{k-j}(1 - \eta_i)\eta_j\|\mathbb{E}[\epsilon_j]\| \tag{24}$$

$$\le \prod_{j=0}^k(1 - \eta_j)\|\Delta\| \tag{25}$$

**Momentum SGD:** For momentum SGD, we have

$$\mathbf{x}_{k+1} = \mathbf{x}_k - \eta_k(\mathbf{x}_k - \sigma_k\epsilon_k) + \gamma(\mathbf{x}_k - \mathbf{x}_{k-1}). \tag{26}$$

Hence, the for the optimum mean $\mathbf{x}^* = 0$, we have

$$\mathbf{x}_{k+1} - \mathbf{x}^* = (1 - \eta_k + \gamma)(\mathbf{x}_k - \mathbf{x}^*) - \gamma(\mathbf{x}_{k-1} - \mathbf{x}^*) + \eta_k \sigma_k \epsilon_k. \tag{27}$$

Let $A_k = \begin{bmatrix} (1 - \eta_k + \gamma)\mathbf{I} & -\gamma\mathbf{I} \\ \mathbf{I} & \mathbf{0} \end{bmatrix}$. Then the above equation can be rewritten into a matrix form:

$$\begin{bmatrix} \mathbf{x}_{k+1} - \mathbf{x}^* \\ \mathbf{x}_k - \mathbf{x}^* \end{bmatrix} = A_k \begin{bmatrix} \mathbf{x}_k - \mathbf{x}^* \\ \mathbf{x}_{k-1} - \mathbf{x}^* \end{bmatrix} + \eta_k \sigma_k \begin{bmatrix} \epsilon_k \\ \mathbf{0} \end{bmatrix} \tag{28}$$

$$= \prod_{j=0}^{k} A_j \begin{bmatrix} \mathbf{x}_1 - \mathbf{x}^* \\ \mathbf{x}_0 - \mathbf{x}^* \end{bmatrix} + \sum_{j=0}^{k-1} \prod_{i=1}^{k-j} A_i \begin{bmatrix} \mathbf{1} \\ \mathbf{0} \end{bmatrix} \sigma_j \eta_j \epsilon_j. \tag{29}$$

Let $\gamma = (1 - \sqrt{\eta_k})^2$, we have $\lambda_{\max}(A_k) = 1 - \sqrt{\eta_k}$, hence $\|A_k\| \leq 1 - \sqrt{\eta_k} < 1$. Denote the initial error as $\Delta = \left\| \begin{bmatrix} \mathbf{x}_1 - \mathbf{x}^* \\ \mathbf{x}_0 - \mathbf{x}^* \end{bmatrix} \right\|$, we derive the norm of expectation error:

$$\|\mathbb{E}[\mathbf{x}_{k+1} - \mathbf{x}^*]\| \leq \left\| \prod_{j=0}^{k} A_j \begin{bmatrix} \mathbf{x}_1 - \mathbf{x}^* \\ \mathbf{x}_0 - \mathbf{x}^* \end{bmatrix} + \mathbb{E}\left[ \sum_{j=0}^{k-1} \prod_{i=1}^{k-j} A_i \begin{bmatrix} \mathbf{1} \\ \mathbf{0} \end{bmatrix} \sigma_j \eta_j \epsilon_j \right] \right\| \tag{30}$$

$$\leq \prod_{j=0}^{k} (1 - \sqrt{\eta_j}) \|\Delta\| + \sigma \sum_{j=0}^{k-1} \left\| \prod_{i=1}^{k-j} A_i \begin{bmatrix} \mathbf{1} \\ \mathbf{0} \end{bmatrix} \right\| \|\mathbb{E}[\eta_j \epsilon_j]\| \tag{31}$$

$$\leq \prod_{j=0}^{k} (1 - \sqrt{\eta_j}) \|\Delta\| \tag{32}$$

When comparing the expectation error $\|\mathbb{E}[\mathbf{x}_k - \mathbf{x}^*]\|$ at $k$-th iteration, we prove that momentum SGD converges faster than vanilla SGD w.r.t. the expectation. □

## D    PROOF OF THEOREM 2

**Theorem 2.** *For any $\delta \in (0, 4\sqrt{1 - \sqrt{\alpha}})$ with $\alpha = 1 - \beta$, if $\gamma = 2 - 2\sqrt{1 - \sqrt{\alpha}} - \sqrt{\alpha} + \delta$, then*

$$\mathbf{x}_T = \zeta_T \mathbf{x}_0 + \kappa_T \epsilon + \gamma \sum_{t=2}^{T-1} \alpha^{\frac{T-t-1}{2}} (\sigma_t - \sigma_{t-1}), \quad \textit{(momentum-based diffusion process)} \tag{33}$$

$$\tilde{\mathbf{x}}_T = \sqrt{\alpha^T} \mathbf{x}_0 + \sqrt{1 - \alpha^T} \epsilon, \qquad \textit{(vanilla diffusion process)} \tag{34}$$

*where $\epsilon \sim \mathcal{N}(0, \mathbf{I})$, $\zeta_T = O(\sqrt{\gamma^T})$, $\kappa_T = \sqrt{1 - \alpha^T + \gamma^2 \alpha^{T-2}(1 - \alpha)}$, $\mathbf{x}_0$ is a starting point shared by both $\mathbf{x}_T$ and $\tilde{\mathbf{x}}_T$.*

*Proof.* From the definition,

$$\mathbf{x}_{t+1} = \sqrt{1 - \beta} \mathbf{x}_t + \sqrt{\beta} \epsilon_t + \gamma(\mathbf{x}_t - \mathbf{x}_{t-1})$$
$$= (\sqrt{1 - \beta} + \gamma)\mathbf{x}_t - \gamma \mathbf{x}_{t-1} + \sqrt{\beta} \epsilon_t$$

Since there is no interaction among different coordinates in this system, we can write this equivalently by

$$(\mathbf{x}_{t+1})_i = (\sqrt{1 - \beta} + \gamma)(\mathbf{x}_t)_i - \gamma(\mathbf{x}_{t-1})_i + \sqrt{\beta}(\epsilon_t)_i$$

and analyze the evolution of each coordinate $i = 1, \ldots, d$ separately. Let $i \in \{1, 2, \ldots, d\}$ and define $x_t = (\mathbf{x}_t)_i$ and $\varepsilon_t = (\epsilon_t)_i$. Then, since $x_{t+1} = (\sqrt{1 - \beta} + \gamma)x_t - \gamma x_{t-1} + \sqrt{\beta}\varepsilon_t$, we have

$$\begin{bmatrix} x_{t+1} \\ x_t \end{bmatrix} = M \begin{bmatrix} x_t \\ x_{t-1} \end{bmatrix} + \sqrt{\beta}\varepsilon_t \begin{bmatrix} 1 \\ 0 \end{bmatrix}$$

$$= M \left( M \begin{bmatrix} x_{t-1} \\ x_{t-2} \end{bmatrix} + \sqrt{\beta}\varepsilon_{t-1} \begin{bmatrix} 1 \\ 0 \end{bmatrix} \right) + \sqrt{\beta}\varepsilon_t \begin{bmatrix} 1 \\ 0 \end{bmatrix}$$

where

$$M = \begin{bmatrix} \sqrt{1-\beta} + \gamma & -\gamma \\ 1 & 0 \end{bmatrix}.$$

Then by induction,

$$\begin{bmatrix} x_{t+1} \\ x_t \end{bmatrix} = M^t \begin{bmatrix} x_1 \\ x_0 \end{bmatrix} + \sum_{k=1}^{t} \sqrt{\beta}\varepsilon_k M^{t-k} \begin{bmatrix} 1 \\ 0 \end{bmatrix}, \tag{35}$$

Since $x_1 = \sqrt{\alpha}x_0 + \sqrt{\beta}\varepsilon_0$,

$$\begin{bmatrix} x_{t+1} \\ x_t \end{bmatrix} = M^t \begin{bmatrix} \sqrt{\alpha}x_0 \\ x_0 \end{bmatrix} + \sqrt{\beta}\varepsilon_0 M^t \begin{bmatrix} 1 \\ 0 \end{bmatrix} + \sum_{k=1}^{t} \sqrt{\beta}\varepsilon_k M^{t-k} \begin{bmatrix} 1 \\ 0 \end{bmatrix}, \tag{36}$$

By solving the characteristic polynomial of $M$, the eigenvalues $\lambda_1, \lambda_2$ of $M$ are

$$\lambda_1 = \frac{1}{2}\left(\sqrt{1-\beta} + \gamma - \sqrt{(\sqrt{1-\beta}+\gamma)^2 - 4\gamma}\right)$$

$$\lambda_2 = \frac{1}{2}\left(\sqrt{1-\beta} + \gamma + \sqrt{(\sqrt{1-\beta}+\gamma)^2 - 4\gamma}\right).$$

Thus, if $(\sqrt{1-\beta}+\gamma)^2 - 4\gamma < 0$, then the eigenvalues are complex and complex conjugates of each other with the same absolute value, which implies that:

$$|\lambda_1|^2 = |\lambda_2|^2 = \lambda_2^*\lambda_2 = \lambda_1\lambda_2 = \det(M_t) = \gamma.$$

where the last line follows from the fact that the product of eigenvalues of a matrix are the determinant of the matrix. We now show that the condition $(\sqrt{1-\beta}+\gamma)^2 - 4\gamma < 0$ is satisfied by our choice of $\gamma = 2 - 2\sqrt{1-c} - c + \delta$ with $\delta \in (0, 4\sqrt{1-c})$ where $c = \sqrt{1-\beta}$: assuming that $\delta > 0$,

$$(\sqrt{1-\beta}+\gamma)^2 - 4\gamma < 0$$
$$\Longleftrightarrow (c + 2 - 2\sqrt{1-c} - c + \delta)^2 < 4(2 - 2\sqrt{1-c} - c + \delta)$$
$$\Longleftrightarrow 4 + 4(1-c) - 8\sqrt{1-c} + \delta^2 + 2\delta(2 - 2\sqrt{1-c}) < 8 - 8\sqrt{1-c} - 4c + 4\delta$$
$$\Longleftrightarrow \delta^2 + 2\delta(2 - 2\sqrt{1-c}) < 4\delta$$
$$\Longleftrightarrow \delta < 4\sqrt{1-c}.$$

Thus, the condition $(\sqrt{1-\beta}+\gamma)^2 - 4\gamma < 0$ is satisfied, implying that $|\lambda_1|^2 = |\lambda_2|^2 = \gamma$. We write its eigendecomposition by $M = Q\Lambda Q^{-1}$, with which

$$\|M^t\| = \|Q\Lambda^t Q^{-1}\| = O(\sqrt{\gamma^t}).$$

For the noise part, we denote the accumulated total noise at $t$ of the coordinate $i$ by $J_t = (\sigma_t)_i$:

$$\left(\sqrt{\beta}\varepsilon_0 M^t \begin{bmatrix} 1 \\ 0 \end{bmatrix} + \sum_{k=1}^{t} \sqrt{\beta}\varepsilon_k M^{t-k)} \begin{bmatrix} 1 \\ 0 \end{bmatrix}\right)_1 = J_{t+1} = \sqrt{1-\beta}J_t + \sqrt{\beta}\varepsilon_t + \gamma(J_t - J_{t-1})$$

$$= \sqrt{1-\beta}(\sqrt{1-\beta}J_{t-1} + \varphi_{t-1}) + \varphi_t$$

$$= \sqrt{\alpha^t}J_1 + \sum_{k=1}^{t}\left((1-\beta)^{\frac{t-k}{2}}\right)\varphi_k$$

$$= \sqrt{\alpha^t}\sqrt{1-\alpha}\varepsilon_0 + \sum_{k=1}^{t}\left((1-\beta)^{\frac{t-k}{2}}\right)\varphi_k$$

where $\varphi_t = \sqrt{\beta}\varepsilon_t + \gamma(J_t - J_{t-1})$. By expanding this $\varphi_t$,

$$J_{t+1} = \sqrt{\alpha^t}\sqrt{1-\alpha}\varepsilon_0 + \sum_{k=1}^{t}\left((1-\beta)^{\frac{t-k}{2}}\right)\sqrt{\beta}\varepsilon_k + \sum_{k=1}^{t}\left((1-\beta)^{\frac{t-k}{2}}\right)\gamma(J_k - J_{k-1}).$$

For the second term, with $\alpha = 1 - \beta$, we use the property of Gaussian random variable as

$$\sqrt{\alpha^t}\sqrt{1-\alpha}\varepsilon_0 + \sum_{k=1}^{t}\left(\alpha^{\frac{t-k}{2}}\right)\sqrt{1-\alpha}\varepsilon_k = \sum_{k=0}^{t}\tilde{\varepsilon}_k,$$

where $\tilde{\varepsilon}_k \sim \mathcal{N}(0, \sigma_k^2)$ and

$$\sigma_k^2 = \left(\alpha^{\frac{t-k}{2}}\right)^2(\sqrt{1-\alpha})^2 = \alpha^{t-k}(1-\alpha) = \alpha^{t-k} - \alpha^{t-k+1}.$$

Since the sum of Gaussian is Gaussian with sum of their variances,

$$\sum_{k=0}^{t}\tilde{\varepsilon} = \hat{\varepsilon},$$

where $\hat{\varepsilon} \sim \mathcal{N}(0, \sigma^2)$ and

$$\sigma^2 = \sum_{k=0}^{t}\left(\alpha^{t-k} - \alpha^{t-k+1}\right) = 1 - \alpha^{t+1}.$$

Since $\hat{\varepsilon} = \sqrt{1-a^{t+1}}\varepsilon$ with $\breve{\varepsilon} \sim \mathcal{N}(0,1)$,

$$J_{t+1} = \sqrt{1-\alpha^{t+1}}\breve{\varepsilon} + \sum_{k=1}^{t}\left((1-\beta)^{\frac{t-k}{2}}\right)\gamma(J_k - J_{k-1}).$$

Combining with (36),

$$x_{t+1} = \begin{bmatrix} x_{t+1} \\ x_t \end{bmatrix}_1 = \left(M^t\begin{bmatrix}\sqrt{\alpha}x_0 \\ x_0\end{bmatrix} + \sqrt{\beta}\varepsilon_0 M^t\begin{bmatrix}1 \\ 0\end{bmatrix} + \sum_{k=1}^{t}\sqrt{\beta}\varepsilon_k\left(M^{t-k}\right)\begin{bmatrix}1 \\ 0\end{bmatrix}\right)_1$$

$$= \left(M^t\begin{bmatrix}\sqrt{\alpha}x_0 \\ x_0\end{bmatrix}\right)_1 + \sqrt{1-\alpha^{t+1}}\breve{\varepsilon} + \sum_{k=1}^{t}\left((1-\beta)^{\frac{t-k}{2}}\right)\gamma(J_k - J_{k-1}).$$

This implies that

$$x_T = \zeta_T x_0 + \sqrt{1-\alpha^T}\breve{\varepsilon} + \sum_{t=1}^{T-1}(1-\beta)^{\frac{T-t-1}{2}}\gamma(J_t - J_{t-1}),$$

where $\zeta_t = O(\sqrt{\gamma^t})$. Moreover, since $J_1 - J_0 = \sqrt{1-\alpha}\varepsilon_0$,

$$\sqrt{1-\alpha^T}\breve{\varepsilon} + (1-\beta)^{\frac{T-1-1}{2}}\gamma(J_1 - J_0) = \sqrt{1-\alpha^T}\breve{\varepsilon} + \sqrt{\alpha^{T-2}}\gamma\sqrt{1-\alpha}\varepsilon_0$$

$$= \sqrt{1-\alpha^T + \gamma^2\alpha^{T-2}(1-\alpha)}\varepsilon$$

where the last line follows from the fact that the sum of two Gaussian is Gaussian with the sum of their variances. Thus,

$$x_T = \zeta_T x_0 + \sqrt{1-\alpha^T + \gamma^2\alpha^{T-2}(1-\alpha)}\varepsilon + \sum_{t=2}^{T-1}(1-\beta)^{\frac{T-t-1}{2}}\gamma(J_t - J_{t-1}),$$

By recalling the definition of $x_t = (\mathbf{x}_t)_i$, $\varepsilon_t = (\epsilon_t)_i$, and $J_t = (E_t)_i$, since $i \in \{1, 2, \ldots, d\}$ was arbitrary, this holds for all $i \in \{1, 2, \ldots, d\}$, implying that

$$\mathbf{x}_{T+1} = \begin{bmatrix}(\mathbf{x}_{T+1})_1 \\ \vdots \\ (\mathbf{x}_{T+1})_d\end{bmatrix}$$

$$= \begin{bmatrix}\zeta_T(\mathbf{x}_0)_1 + \sqrt{1-\alpha^T + \gamma^2\alpha^{T-2}(1-\alpha)}(\epsilon)_1 + \sum_{t=2}^{T-1}(1-\beta)^{\frac{T-t-1}{2}}\gamma((E_t)_1 - (E_{t-1})_1) \\ \vdots \\ \zeta_T(\mathbf{x}_0)_d + \sqrt{1-\alpha^T + \gamma^2\alpha^{T-2}(1-\alpha)}(\epsilon)_d + \sum_{t=2}^{T-1}(1-\beta)^{\frac{T-t-1}{2}}\gamma((E_t)_d - (E_{t-1})_d)\end{bmatrix}$$

$$= \zeta_T\mathbf{x}_0 + \sqrt{1-\alpha^T + \gamma^2\alpha^{T-2}(1-\alpha)}\epsilon + \sum_{t=2}^{T-1}(1-\beta)^{\frac{T-t-1}{2}}\gamma(E_t - E_{t-1}),$$

where $\zeta_T = O(\sqrt{\gamma^T})$.

$\square$

