# OpenReview forum: "Momentum-accelerated Diffusion Process for Faster Training and Sampling"
_ICLR.cc/2024/Conference — Submitted to ICLR 2024_

### Official Review · Reviewer_LPzE · 2023-10-31

**Soundness:** 2 fair
**Presentation:** 2 fair
**Contribution:** 2 fair
**Rating:** 5
**Confidence:** 4

**Summary:**

This work presents a momentum-accelerated diffusion model for faster training and sampling. Empirical results are reported by applying the proposed FDM into several diffusion models (VP, VE, EDM). Three datasets (CIFAR-10, FFHQ, and AFHQv2) are used for evaluation.

**Strengths:**

- It is good to evaluate the proposal under different diffusion models.

- Detailed theoretical analysis is provided.

**Weaknesses:**

- The idea to connect SGD with forward process of diffusion process is interesting. However, there's a main difference between them: the gradient $g_t$ in SGD has unique relation to $x_t$ but $\epsilon_t$ is only a random variable, and is independent with $x_t$. They only share the formulation of the formula but has few connection in their meaning. I doubt under this situation, the momentum in the forward process still makes sense.

- 'following EDM, we remove the stochastic gradient noise in Eq. (2)'. Could you point out where EDM has stated this? Discarding stochasticity during forward process seems strange. If there's no stochasticity in forward process, then the equation makes no sense but only $x_T = \prod_{i=1}^{T} \alpha_t x_0$ and there's nothing for the model to learn.

- All the three datasets are somewhat small. It is necessary to evaluate FDM on large-scale dataset with higher resolution (like ImageNet 256x256).

- It is better to show more results by applying FDM into state-of-the-art diffusion model (Stable Diffusion).

**Questions:**

Please check the details in Weaknesses section

---

> ### Author Response · Authors · 2023-11-20
>
> Thank you for your valuable and insightful comments.  In the following, we provide our point-by-point response and hope our response helps address your concerns. We also look forward to the subsequent discussion which may further help solve the current issues.
>
> **Question 1: The idea to connect SGD with forward process of diffusion process is interesting. However, there's a main difference between them: the gradient $g_t$ in SGD has unique relation to $x_t$ but $\epsilon_t$ is only a random variable, and is independent with $x_t$. They only share the formulation of the formula but has few connection in their meaning. I doubt under this situation, the momentum in the forward process still makes sense.**
>
> **Response 1:**
> Since we are not certain that we fully understand your question, we will answer it in two aspects, and hope to address your concern.
>
> **(1) SGD on our constructed dynamic function share the exactly same formulation with the forward diffusion process.** On the constructed dynamic function $f(x) = \mathbb E_{\zeta \sim \mathcal{N}(0, bI)} \frac{1}{2} \lVert x - \frac{\beta_t}{1 - \alpha_t} \zeta \rVert_2^2$, SGD first samples a minibatch data {$\zeta_k$}, and then  computes stochastic gradient  $g_t = x_t - \frac{\beta_t}{b(1 - \alpha_t)} \sum\nolimits_{k=1}^b \zeta_k= x_t - \frac{\beta_t}{1 - \alpha_t} \epsilon_t$ where we use re-parameterization trick by replacing $\frac{1}{b} \sum\nolimits_{k=1}^b \zeta_k \sim \mathcal{N}(0, I)$ with $\epsilon_t \sim \mathcal{N}(0, I)$. Finally, SGD updates parameter $x$ via the following stochastic optimization process:
> $$
>     x_{t+1} =  x_t - (1 - \alpha_t) g_t  =  \alpha_t x_t + \beta_{t} \epsilon_t,
> \tag{1}
> $$
> where learning rate $(1 - \alpha_t) > 0$ is to align the diffusion and SGD processes. Above Eq. (1) exactly aligns with the forward diffusion process in DDPM.
>
> From the above steps, we can find that the stochastic  gradient $g_t = x_t - \frac{\beta_t}{b(1 - \alpha_t)} \sum\nolimits_{k=1}^b \zeta_k= x_t - \frac{\beta_t}{1 - \alpha_t} \epsilon_t$ in SGD involves the parameter $x_t$ and the noise $\beta_{t} \epsilon_t$. But the noise $\beta_{t} \epsilon_t$ is induced from the minibatch sample {$\zeta_k$}.  More importantly, the optimization works, e.g., [1,2], does not care the specific formulation of  stochastic gradient, such as whether stochastic gradient $g_t$ includes the parameter  $x_t$ and an independent noise $\beta_{t}\epsilon_t$, it only requires whether the stochastic  gradient $g_t$ is an unbiased estimation to the full gradient of an objective $f(x)$. Then these works can prove faster convergence speed of momentum SGD than vanilla SGD on  the traditional strongly convex problems.  For our  dynamic function, our Theorem 1 also shows faster convergence speed of momentum SGD than vanilla SGD.
>
> Besides, our momentum forward diffusion process can be applied to several popular diffusion frameworks, e.g. VP, VE, and EDM, and reduces their training cost by about 50% and inference cost by 66% with comparable image synthesis performance across several datasets. These practical results show the effectiveness of the momentum forward diffusion process.
>
> **(2) The roles of $g_t$, $x_t$ and $\epsilon_t$ in the SGD and forward process are the same.**
>
> + In SGD: $g_t$ represents the stochastic gradient used to update $x_t$. This gradient is a function of $x_t$ itself and the noise term $\epsilon_t$ due to the randomness of sampling a minibatch of data {$\zeta_k$}. The noise term $\epsilon_t$ in SGD is essential to simulate the stochastic nature of gradient estimation, please refer to **(1)** for details.
>
> + In the forward process: The analogy in the diffusion process lies in how $x_t$ and $\epsilon_t$ interact to drive the process. The noise term in the diffusion process plays a similar role to that in SGD, introducing randomness that perturbs the process. In this context, $g_t$ can be seen as an analogous mechanism that influences the evolution of $x_t$ over time, although the specific dynamics and objectives differ from SGD.
>
> The key point of similarity lies in the stochastic nature of both processes. In SGD, the stochasticity helps in escaping local minima and exploring the solution space, whereas in the diffusion process, it accelerates the transition from a data distribution to a Gaussian distribution. This parallel in their mechanisms underlines the relevance of using momentum in both contexts. Momentum, by accumulating historical updates (in SGD) or influencing the trajectory (in the diffusion process), helps in smoothing out the updates and providing a more stable and efficient trajectory towards the objective, whether it be finding the optimum of a function or effectively transitioning through the diffusion process.
>
> [1] Christopher De Sa, https://www.cs.cornell.edu/courses/cs4787/2020sp/lectures/Lecture7.pdf
>
> [2] Yanli Liu, Yuan Gao, Wotao Yin, An Improved Analysis of Stochastic Gradient Descent with Momentum, NeurIPS 2020.

---

> ### Author Response · Authors · 2023-11-20
>
> **Question 2: 'following EDM, we remove the stochastic gradient noise in Eq. (2)' Could you point out where EDM has stated this?**
>
> **Response 2:** Thanks. For EDM (see arXiv.2206.00364), it claims this at least two places.
> On page 4, it states "The main purpose of this reframing is to bring into light all the independent components that often appear tangled together in previous work. In our framework, there are no implicit dependencies between the components -- any choices (within reason) for the individual formulas will, in principle, lead to a functioning model."
>
> On page 21, it also says "Given that the idea of the probability flow ODE is to match a particular set of marginal distributions, it makes sense to treat the marginal distributions as first-class citizens and define the ODE directly based on $\sigma(t)$ and $s(t)$, eliminating the need for $f(t)$ and $g(t)$."
>
> Please see more details in the corresponding sections. The EDM design space eliminates implicit dependencies between each component, including mean scaling $\mu_t$ and noise schedule $\sigma_t$. This independence grants us sufficient flexibility to separately design the sample mean scaling and noise accumulation. Inspired by this, we derive the expectation term of our momentum-based diffusion process, as elaborated from Eq. (9) to Eq. (13) in our paper. In this part of the derivation, the noise term is notably absent, as the change in expectation is not dependent on the noise term.
>
>
> **Question 3: Discarding stochasticity during forward process seems strange. If there's no stochasticity in forward process, then the equation makes no sense but only $x_T = \prod_{t=1}^T \alpha_t x_0$ and there's nothing for the model to learn.**
>
> **Response 3:**
> **Our momentum-based forward process indeed incorporates stochastic noise.** This is explicitly demonstrated in the formulation of the perturbation kernel in Eq. (14) of our paper. For certain parts of our derivation, it do not involve the noise term, since we aim to derive the expectation term of our momentum-based diffusion process. For this,  please refer to Response 2 for detailed elaboration. After deriving the expectation term,   we integrate the noise term into our perturbation kernel later in Eq. (14) and Eq. (15).  For instance, according to Eq. (15), we sample the noisy sample at time step $t$ by $x_t = e^{-\mathcal{B}(t)}(1 + \mathcal{B}(t))x_0 + \sigma_t \epsilon_t$, where $\epsilon_t \sim \mathcal{N}(0, I)$. This term $\epsilon_t$ represents stochastic noise and is integral to the forward process, ensuring that stochasticity is maintained.
>
> **Question 4: All the three datasets are somewhat small. It is necessary to evaluate FDM on large-scale dataset with higher resolution (like ImageNet 256x256).**
>
> **Response 4:**
> **Our experimental approach has been evaluated on several widely used datasets.** It is comparable in scale to those used in other papers such as [3-5]. Conducting experiments on these datasets required significant computational resources, specifically 8 A100 GPUs for approximately 4 days of training. This level of investment in terms of hardware and time underscores our commitment to rigorous testing and validation of our FDM.
>
> We acknowledge that ImageNet 256x256 represents a larger-scale and higher-resolution dataset that could further test the capability of our model. However, it is important to note that ImageNet 256x256 is not typically used as a standard dataset for pixel-level diffusion models due to the high amount of GPU resource requirements. See the experiments in VP, VE, and EDM to check.
> Moreover, even training on a dataset like ImageNet 64x64 would require substantial resources, specifically 32 A100 GPUs for about 14 days, which is beyond our current GPU resource capacity.
>
> [3] Dockhorn, Tim, Arash Vahdat, and Karsten Kreis. Score-Based Generative Modeling with Critically-Damped Langevin Diffusion. ICLR 2022.
>
> [4] Zhang, Qinsheng, Molei Tao, and Yongxin Chen. gDDIM: Generalized Denoising Diffusion Implicit Models. ICLR 2023.
>
> [5] Du, Weitao, He Zhang, Tao Yang, and Yuanqi Du. A flexible diffusion model. ICML 2023.

---

> > ### Author Response · Authors · 2023-11-20
> >
> > **Question 5: It is better to show more results by applying FDM into state-of-the-art diffusion model (Stable Diffusion).**
> >
> > **Response 5:**
> > **The application of our FDM to Stable Diffusion was considered but found to be impractical due to very high training cost and unavailability of official and completed training codebase of Stable Diffusion.**  For training cost, training Stable Diffusion from scratch requires 256 A100 GPUs on Amazon Web Services for a total of 150,000 GPU-hours, at a cost of \$600,000 [6]. Such a level of resource investment is beyond the scope and feasibility of our current research efforts. Additionally, there is a challenge regarding the availability of complete training code from the open-source community. Details like computational complexity and training instructions are often not fully disclosed, which poses a significant obstacle for reproducing Stable Diffusion.
> >
> > Besides, our research primarily aims at improving the efficiency of diffusion models. In pursuit of this goal, we have conducted comprehensive experiments using various popular DMs, including VP, VE and EDM. Notably, the EDM is already recognized as a SoTA diffusion model within our experimental scope. These experiments have been rigorous and effectively demonstrate the significant improvements and effectiveness of our FDM.
> >
> > [6] https://huggingface.co/CompVis/stable-diffusion-v1-4

---

> ### Author Response · Authors · 2023-11-21
>
> Dear Reviewer LPzE,
>
> Thank you for your insightful and valuable comments, which have been of great help to us. The deadline for our discussion is approaching. If you have any further concerns or questions, welcome to discuss them with us.
>
> Moreover, another two reviewers have positive comments on this work. So could you also read their comments and decide whether you change the score? Many thanks.
>
> Bests, Authors

---

> > ### Comment · Reviewer_LPzE · 2023-11-22
> > **Feedback**
> >
> > Thanks for the authors' response.
> >
> > Response 1 has addressed my doubts, I still have some doubts about Response 2&3 though.
> >
> > For Response 2: The first place in EDM provided by the authors says to disentangle different design principles, but not remove the stochastic terms in forward process; while the second place is about reverse process. But the paper actually discusses forward process around Eq. (9) and the two processes are different. Combining with Response 3, my point is that: "it's inappropriate to separate the noise in Eq. (9), deriving Eq. (10) - Eq. (12) without the stochastic gradient noise, and add it back later". Like the statement in Response 1, "$x_t$ and $\epsilon_t$ interact to drive the process."

---

> ### Author Response · Authors · 2023-11-22
>
> We are grateful for your insightful comments. We hope that our response will be satisfactory and, if so, would you be willing to increase your score?
>
> **Our approach is valid from both the perspective of designing in EDM framework and solving a SDE.**
> The first place of EDM, it emphasizes that each component within the design is distinct and independent. In the second place of EDM implies that the expectation term $\mu(t)$ and variance term $\sigma(t)$ have distinct effects on the reverse process. Together, these two aspects provide assurance that the expectation term and variance term can be defined separately.
>
> On the other hand, from the Stochastic Differential Equation (SDE) perspective, as evidenced in [1], particularly on page 273, Eq. (12.76).
> For a SDE process:
> $$
> \mathrm{d} \mathbf{x}= f(t) \mathbf{x}\mathrm{d} t + g(t) \mathrm{d} \boldsymbol{w},
> $$
> the corrosponding solution $p_t(\mathbf{x}(t))) = \mathcal{N}(\mu(t), \sigma^2(t))$ is defined as:
> $$
> \begin{align}
> & \frac{\mathrm{d} \mu(t)}{\mathrm{d} t}=f(t) \mu(t) \\\\
> & \frac{\mathrm{d} \sigma(t)}{\mathrm{d} t}=f(t) \sigma(t)+\sigma(t) f(t)^{\top}+g(t) g(t)^{\top}.
> \end{align}
> $$
> It is clear that solving the mean-varying process does not depend on the variance term. This is further supported by Eq. (32) in our Appendix, where we demonstrate that the expectation is not dependent on the variance.
>
> The EDM framework not only validates our approach to defining a mean-varying process without incorporating noise but also **enables us to directly adopt the variance term from corresponding diffusion models without any modification**. This approach aligns with the EDM principle on page 4 in EDM paper: *"In other words, changing one component does not necessitate changes elsewhere in order to, e.g., maintain the property that the model converges to the data in the limit. In practice, some choices and combinations will of course work better than others."*
>
> In our paper, we highlight this approach as *"incorporating the expectation of momentum into the vanilla diffusion process of a certain DM."* This method allows for a seamless integration of our momentum-based diffusion process with existing DMs, leveraging the independent nature of expectation and variance terms within the EDM framework.
>
> Therefore, our approach maintains the integrity and independence of the variance term in correspondence with the DMs while effectively integrating the momentum expectation into the diffusion process, ensuring a balanced and effective diffusion model design.
>
> [1] Särkkä, Simo, and Arno Solin. Applied stochastic differential equations. Vol. 10. Cambridge University Press, 2019.

---

> > ### Comment · Reviewer_LPzE · 2023-11-22
> > **Feedback**
> >
> > Thanks for the response. I agree with the author's claim that in reverse process solving the mean-varying process does not depend on the variance term. Actually, in EDM, the independence of the change of mean and variance in forward process have not been discussed. Nevertheless, in this paper, the momentum technique is implemented in forward process. So it is better to provide more evidence on the similar claims in forward process.

---

> ### Author Response · Authors · 2023-11-22
>
> Thank you for acknowledging our explanation regarding the independence of mean and variance changes in the reverse process. We appreciate your further inquiry about the forward process and are glad to provide more clarification.
>
> **In the EDM framework, it is explicitly defined that the mean and variance can vary independently in the forward process.** This principle is outlined in the EDM paper, stating: *"In our framework, there are no implicit dependencies between the components."* To elucidate this concept clearly, let us compare the forward process in EDM and traditional SDE-based formulations [1].
>
> In [1], the forward process is defined by an SDE:
> $$
> \mathrm{d} \mathbf{x}=\mathbf{f}(t) \mathbf{x} \mathrm{d} t+g(t) \mathrm{d} \mathbf{w},
> $$
> and the reverse ODE is subsequently derived as:
> $$
> \mathrm{d} \mathbf{x}=\left[\mathbf{f}(t)\mathbf{x}- \frac{1}{2} g(t)^2 \nabla_{\mathbf{x}} \log p_t(\mathbf{x})\right] \mathrm{d} t.
> $$
> The perturbation kernel $p_t(\mathbf{x}_t | \mathbf{x}_0)$ which is required for both training and sampling is then formulated based on above forward SDE, formulated by specific functions $f(t)$ and $g(t)$ (see Eq. (11) in EDM Appendix B.1).
>
> **On the contrary, the EDM framework takes a direct approach to defining the critical perturbation kernel representing the forward process:**
> $$
> p_t(\mathbf x_t|\mathbf x_0)=\mathcal{N}(\mathbf x_t;\mu(t)\mathbf x_0,\mu(t)^2\sigma(t)^2\mathbf{I}),
> $$
> **Here, the mean-varying $\mu(t)$ and noise accumulation $\sigma(t)$ are defined separately.** The corresponding reverse process is then derived based on $\mu(t)$ and $\sigma(t)$:
> $$
> \mathrm{d} \mathbf{x}=\left[\frac{\dot{\mu}(t)}{\mu(t)} \mathbf{x}-\mu(t)^2 \dot{\sigma}(t) \sigma(t) \nabla_{\mathbf{x}} \log p\left(\frac{\mathbf{x}}{\mu(t)} ; \sigma(t)\right)\right] \mathrm{d} t.
> $$
> This approach eliminates the need to explicitly define a forward SDE, which is supported by evidence from the EDM paper page 21: *"Given that the idea of the probability flow ODE is to match a particular set of marginal distributions, it makes sense to treat the marginal distributions as first-class citizens and define the ODE directly based on $\sigma(t)$ and $\mu(t)$, eliminating the need for $f(t)$ and $g(t)$."*
>
> **This distinct approach within the EDM framework ensures the independence of each component, including $\mu(t)$ and $\sigma(t)$, in both forward and reverse processes.** This foundational independence within the framework effectively supports the implementation of our momentum technique in the forward process.
>
> [1] Song, Yang, Jascha Sohl-Dickstein, Diederik P. Kingma, Abhishek Kumar, Stefano Ermon, and Ben Poole. Score-Based Generative Modeling through Stochastic Differential Equations. ICLR 2021

---

> ### Author Response · Authors · 2023-11-23
>
> Dear Reviewer LPzE,
>
> We are grateful for your insightful comments and appreciate the opportunity to clarify our work further. We hope that our responses have adequately addressed your concerns. If you find our explanations satisfactory, would you be willing to increase your score? If you have any further concerns or questions, welcome to discuss them with us.
>
> Bests, Authors

---

> > ### Comment · Reviewer_LPzE · 2023-11-23
> >
> > Thanks again for your detailed response, which has addressed my concerns. I will increase my rating.

---

> > > ### Author Response · Authors · 2023-11-23
> > >
> > > Dear Reviewer LPzE:
> > >
> > > We are pleased to hear that our rebuttal successfully addressed your concerns!
> > >
> > > As this is the last day of the discussion period, we would be grateful for any further thoughts or an update to the rating if you feel it would be appropriate. We are here to answer any additional questions that you may have.
> > >
> > > Thank you again! We appreciate you taking the time to review our paper.
> > >
> > > Best,
> > > Authors

---

### Official Review · Reviewer_ZeuL · 2023-10-31

**Soundness:** 3 good
**Presentation:** 3 good
**Contribution:** 3 good
**Rating:** 8
**Confidence:** 3

**Summary:**

This work incorporates heavy-ball momentum into the diffusion process of a diffusion model to speed up its training and inference. Specifically,  it first shows that the forward diffusion process can be viewed as an iterative scheme of stochastic gradient descent (SGD) applied to minimize a time-varying quadratic function. Motivated by this, it adds a heavy-ball-type momentum term to the forward diffusion process. To derive a concrete algorithm, it translates the discrete-time forward process into a (deterministic) critically damped ODE (noise term treated as a constant following EDM (Karras et al. [1])). By solving the ODE, it obtains a perturbation kernel, which is incorporated with other diffusion models and used in the reverse sampling process. The numerical experiments demonstrate that the proposed approach can speed up both training and inference given a certain budget.

[1] Elucidating the Design Space of Diffusion-Based Generative Models

**Strengths:**

- This paper is well-organized and can be easily followed. The idea of introducing momentum into the forward diffusion process seems to be novel and the contributions are made clear.

- The experiments are promising, showing that the integration of the modified forward diffusion process (including momentum) with other models seem to work well, and the improvement in terms of training and inference speed is consistent over the baselines.

**Weaknesses:**

- Some claims are not fully supported. Theorem 1 is mainly a convergence result of SGD with momentum on a quadratic function. It is unclear how this rate would imply a faster convergence speed of the modified forward diffusion process (eqn 2) despite some similarities in the iterates. This is further complicated by the fact that the noise term in eqn 2 is treated as a constant in eqn 9. Theorem 2 shows the convergence of the proposed diffusion process to a Gaussian distribution. However, it does not really quantify the rate of convergence, and its implications on the reverse sampling process is largely a heuristic.

- The actual algorithm (and its implementation) differs from the iterates considered in Theorem 2. Thus, the theory does not really capture the training dynamics.

**Questions:**

Despite some weakness in the theory justifications. The proposed approach seems to open up some new directions in speeding up diffusion process by drawing ideas from the optimization perspectives.

---

> ### Author Response · Authors · 2023-11-20
>
> Thank you for the insightful and positive comments! In the following, we provide our point-by-point response and hope our response helps address your concerns. We also look forward to the subsequent discussion which may further help solve the current issues.
>
> **Question 1: Theorem 1 is mainly a convergence result of SGD with momentum on a quadratic function. It is unclear how this rate would imply a faster convergence speed of the modified forward diffusion process (eqn 2) despite some similarities in the iterates.**
>
> **Response 1:**
> **The convergence rate of the optimization problem we examine is directly analogous to the convergence rate of the diffusion process towards equilibrium.**  This relationship is elucidated through two key points:
>
> **(1) Alignment of Processes:**  According to Section 4.1, we have that, when minimizing a stochastic quadratic time-variant function $f(\mathbf{x}) = \mathbb{E}_{\zeta \sim \mathcal{N}(0, b\mathbf{I})} \frac{1}{2} \lVert \mathbf{x} - \frac{\beta_t}{1 - \alpha_t} \zeta \rVert_2^2$ with learning rate $\eta_t = 1 - \alpha_t$, the SGD optimization process aligns with the diffusion process in each iteration. This alignment establishes a clear link between the dynamics of the SGD and the diffusion process.
>
> **(2) Alignment of Equilibrium State:** The optimal variable for minimizing $f(\mathbf{x})$ can be calculated as $\mathbf{x}^*=0$, which aligns with the expectation of equilibrium state of the diffusion process. According to Theorem 1, the expectation error in momentum SGD decreases more rapidly towards this optimal state. This implies that the momentum-based diffusion process also approaches equilibrium at an accelerated rate.
>
>
> **Question 2: This is further complicated by the fact that the noise term in eqn 2 is treated as a constant in eqn 9.**
>
> **Response 2:**
> **In our derivation, we consistently treat the noise term as a variable, not a constant.** In the EDM design space, the probability flow ODE of the reverse process is defined by two distinct and independent factors: mean scaling $\mu_t$ and noise schedule $\sigma_t$. This independence grants us sufficient flexibility to separately design the sample mean scaling and noise accumulation. We begin by deriving the expectation term of our momentum-based diffusion process, as elaborated from Eq. (9) to Eq. (13) in our paper. In this part of the derivation, the noise term is notably absent, as the change in expectation is not dependent on the noise term.
>
> In subsequent developments of our model, we integrate the noise schedule $\sigma_t$ directly from the relevant baseline DMs as shown in Eq. (14) and Eq. (15). The key here is the independence of $\sigma_t$ from $\mu_t$, which allows us to adopt it unaltered. In practice, we found that numerical computation of the closed-form solution for accumulated noise like CLD does not yield practical benefits. This approach can be interpreted as integrating the expectation of momentum into the vanilla diffusion process of a certain DM. This integration is seamless, facilitating the incorporation of our momentum-based diffusion process with existing DMs, which enhances the existing DMs' performance while preserving their fundamental convergence property to the data distribution. This approach not only maintains the integrity of the baseline models but also enriches them with the efficiency and effectiveness of momentum-based diffusion.

---

> ### Author Response · Authors · 2023-11-20
>
> **Question 3-1: Theorem 2 shows the convergence of the proposed diffusion process to a Gaussian distribution. However, it does not really quantify the rate of convergence.**
>
> **Response 3-1:**
> The rate of convergence is specifically addressed in Theorem 1. In Theorem 1, we demonstrate the faster convergence of our momentum-based diffusion process compared to the vanilla diffusion approach. On the other hand, Theorem 2 focuses on establishing a clear relationship between $x_t$ and $x_0$, whose primary goal is to provide an intuitive understanding of how the momentum-based diffusion process evolves over time, especially to highlight the difference from the vanilla diffusion process. Theorems 1 and 2 together provide a holistic view of our model. While Theorem 1 provides quantitative analysis regarding the convergence speed, Theorem 2 contributes to the understanding of the state transitions in our momentum-based diffusion process.
>
> **Question 3-2: Its implications on the reverse sampling process is largely a heuristic.**
>
> **Response 3-2:**
> **Since the reverse process is decided by its forward diffusion process, one can expect the accelerated speed as the forward one, and also enjoy the acceleration effect.**  This is because our momentum is implicitly incorporated into the score function in the reverse process. For example, $\nabla \log p_{\text{VP}}(x_t | x_0) \propto -\frac{x_t - e^{-\mathcal{B}(t)}x_0}{\beta(t)^2}$ in VP, while $\nabla \log p_{\text{FDM}}(x_t | x_0) \propto -\frac{x_t - e^{-\mathcal{B}(t)}(1 + \mathcal{B}(t))x_0}{\beta(t)^2}$ in our FDM. The only difference comes from the momentum. In these formulations, we can see the acceleration in the forward process results in the acceleration in the reverse process. Intuitively, the forward process in the momentum diffusion process has the formulation $x_{t+1} = x_t + \beta_t \epsilon + \alpha (x_{t}-x_{t-1})$, which leads to a larger distance between $x_{t}$ and $x_{t+1}$ than the one in the vanilla diffusion process. Thus, in the backward process of the momentum diffusion process, using $x_{t+1}$ to predict $x_{t}$ is also a more aggressive prediction. Moreover, our experiments on both EDM sampler (Table 3) and DPM-Solver++ (Table 4) affirm the acceleration on the reverse sampling process in practice.
>
> **Question 4: The actual algorithm (and its implementation) differs from the iterates considered in Theorem 2. Thus, the theory does not really capture the training dynamics.**
>
> **Response 4:**
> **The iteration considered in Theorem 2 is an abstraction for ease of notation and analysis.** In Theorem 2, we  set $\beta_t \equiv \beta$ to streamline our analysis. While the abstraction in Theorem 2 may not capture every nuance of the actual algorithm's implementation, it still faithfully reflects the fundamental mechanism of the diffusion process that gradually diffuses a data point into random noise. By abstracting the iteration process in this way, we aim to provide an intuitive and clear understanding of how the momentum-based diffusion process evolves, especially to highlight the difference from the vanilla diffusion process. This type of abstraction is a key in physics and mathematical theory to gain insights.
>
> Although this approach does not encompass all the complexities of the training dynamics found in practical implementations, it still offers valuable insights. It allows us to identify the fundamental aspects of the diffusion process and the role of momentum in it. The insights gained from this abstracted state transition in Theorem 2 are critical to guide the practical application and development of our FDM.

---

> ### Author Response · Authors · 2023-11-21
>
> Dear Reviewer ZeuL,
>
> Thank you for your insightful and positive comments, which have been of great help to us. The deadline for our discussion is approaching. If you have any further concerns or questions, welcome to discuss them with us.
>
> Bests, Authors

---

> ### Author Response · Authors · 2023-11-23
>
> Dear Reviewer ZeuL:
>
> Thank you for your insightful and positive comments, which have been of great help to us.
>
> As this is the last day of the discussion period, we would be grateful for any further thoughts or an update to the rating if you feel it would be appropriate. We are here to answer any additional questions that you may have.
>
> Thank you again! We appreciate you taking the time to review our paper.
>
> Best, Authors

---

### Official Review · Reviewer_GwvH · 2023-11-01

**Soundness:** 2 fair
**Presentation:** 3 good
**Contribution:** 3 good
**Rating:** 5
**Confidence:** 4

**Summary:**

The paper investigate to incorporate the momentum SGD into the diffusion process and propose a method named Fast Diffusion Model (FDM) to speed up diffusion models. Several experiments are conducted, and the results were compared against existing models to validate the effectiveness of the proposed method.

**Strengths:**

1. The authors not only analyze the proposed method from the theoretical apsect but also validate it empirically. The experimental results seems solid. Also, accelaration of diffusion process is a meaningful problem to investigate.
2. The paper is well-organized, and it is quite easy for readers to follow.

**Weaknesses:**

1. Using momentum to accelerate the optimation seems not a new idea, and the contribution is somewhat limited.
2. The theoritical analyses seem not very rigorous. See Questions part.

**Questions:**

In Theorem 1, the author argues that the momentum SGD is faster by comparing the errors in expectation. However, does the upper bound of SGD tight? It is not safe to draw the conclusion that SGD is slower by simply showing that the upper bound is higher than momentum SGD, epsically when we are not aware whether this upper bound is tight.

---

> ### Author Response · Authors · 2023-11-20
>
> Thank you for your valuable and insightful comments! In the following, we provide our point-by-point response and hope our response helps address your concerns. We also look forward to the subsequent discussion which may further help solve the current issues.
>
>
> **Question 1: Using momentum to accelerate the optimation seems not a new idea, and the contribution is somewhat limited.**
>
> **Response 1:**
> **The incorporation of momentum into the diffusion process, as presented in our work, represents a substantial advancement in this field.** While momentum is a well-known acceleration technique in optimization, only a few works focus on its application within the diffusion process, which we have detailed compared in Section 2 and Section 5.3.
> Our contribution is listed as follows:
>
> **(1) Theoretical Insights:** Our work bridges a crucial gap, establishing a clear and direct link between diffusion processes and gradient descent by constructing a dynamic function.   This connection inspires our exploration of the momentum-accelerated diffusion process. Moreover, we provide theoretical analysis to show the faster convergence rate in Theorem 1,  and establish a clear relationship between the perturbed image $\mathbf{x}_t$ and the clean image $\mathbf{x}_0$ in Theorem 2 which also shows the different convergence speed of momentum-accelerated diffusion process and the vanilla one.
>
> **(2) Practical Insights:** We have effectively demonstrated the seamless integration of a momentum-based diffusion process across various diffusion models. To our knowledge, we are the first to successfully integrate momentum into the diffusion process without necessitating an *additional* momentum (velocity) space. Unlike other momentum-based methods that utilize an auxiliary space to simplify the computation of the perturbation kernel, our approach involves a more challenging task. We aim to directly derive the desired perturbation kernel of the momentum-based diffusion process exclusively within the sample space. To overcome this issue, we follow EDM and eliminate the implicit dependencies between each component, including mean scaling $\mu_t$ and noise schedule $\sigma_t$. This independence grants us sufficient flexibility to separately design the sample mean scaling and noise accumulation, thus allowing us to formulate the forward mean-varying process by a second-order damped-oscillation ODE (Eq. (12)) without the auxiliary velocity term.
> This innovative approach offers numerous benefits such as numerical stability, memory efficiency, and wide applicability, which paves the way for more efficient and streamlined diffusion processes.
>
> **(3) Experimental Results:** Through extensive experiments, we verified improvements in training and sampling efficiency, along with superior image generation quality.
> Specifically, our FDM reduces the training cost of DMs by about 50\% and the inference cost by 66\% with a comparable image synthesis performance across several datasets.
> This practical application of our theoretical insights underscores the real-world impact of our research.
>
> The additional benefit of our approach is evident in our experimental study presented in Table 7. In the experiments, both CLD and our FDM were applied using the traditional RK45 solver. However, a significant distinction was observed when a more efficient EDM sampler was employed. While CLD exhibited a notable performance decrease with the EDM sampler, our FDM maintained consistent performance.
> Moreover, we have extended our experimental analysis to include gDDIM[1], which addresses oscillation/overshoot and numerical instability issues commonly found in the vanilla CLD. The results of these experiments are presented below. Despite the specific enhancements implemented in gDDIM to improve upon CLD, the experimental results demonstrate that our FDM still significantly outperforms gDDIM. This further underscores the robustness and efficiency of our FDM in comparison to previous methods.
>
> **Table: Image synthesis performance on CIFAR-10 with 35 NFEs. gDDIM (CLD) use specifically designed gDDIM sampler, while VP-FDM use EDM sampler. The result denoted with $\dagger$ is from the original paper, with 50 NFEs.**
> |    Method   | 30 Mimg | 50 Mimg | 70 Mimg | 100 Mimg |      Best      |
> |:-----------:|:-------:|:-------:|:-------:|:--------:|:--------------:|
> | gDDIM (CLD) |   5.15  |   2.95  |   2.76  |   2.51   | 2.25$^\dagger$ |
> | VP-FDM      |   4.01  |   2.74  |   2.37  |   2.24   |      2.08      |
>
> [1] Zhang, Qinsheng, Molei Tao, and Yongxin Chen. gDDIM: Generalized Denoising Diffusion Implicit Models. ICLR 2023.

---

> ### Author Response · Authors · 2023-11-20
>
> **Question 2: In Theorem 1, the author argues that the momentum SGD is faster by comparing the errors in expectation. However, does the upper bound of SGD tight? It is not safe to draw the conclusion that SGD is slower by simply showing that the upper bound is higher than momentum SGD, especially when we are not aware whether this upper bound is tight.**
>
> **Response 2:**
> **The bound for vanilla SGD, as presented in Theorem 1, is indeed strict and accurately reflects the convergence behavior.** To elucidate this, we delve into the step-by-step derivation of the expectation error, beginning with Eq. (22) in  our appendix:
> $$
> \begin{align}
> 	\lVert \mathbb{E}[x_{k+1}-x^*] \rVert
> 	&  \overset{(i)}{=} \lVert \prod_{j=0}^k (1 - \eta_j) (x_0 - x^*) + \mathbb{E} [\sum_{j=0}^k \prod_{i=1}^{k-j} (1 - \eta_i) \eta_j \sigma_j \epsilon_j ] \rVert  \\\\
> 	& \overset{(ii)}{=} \lVert  \prod_{j=0}^k (1 - \eta_j)  (x_0 - x^*) \rVert  \\\\
> 	& = \lVert x_0 - x^* \rVert \prod_{j=0}^k (1 - \eta_j) ,
> \end{align}
> $$
>
> where $(i)$ is due to $x_i$ is uncorrelated to $\epsilon_j$ for any i,j, and $(ii)$ is due to $\mathbb{E}[\epsilon_j] = 0$. Similarly, for the forward diffusion process in DDPM, we can derive  $\lVert \mathbb{E}[\prod_{k=1}^t \alpha_k x_0] \rVert = \lVert \prod_{k=1}^t (1 - \eta_k) x_0 \rVert= \lVert x_0 - x^* \rVert \prod_{k=1}^t (1 - \eta_k) $ where we use $\alpha_t = 1 - \eta_t$ and $x^* = 0$. In this way, for both results of the momentum forward process and the vanilla one, their bounds are tight since both use "=". Our Theorem 1 can successfully show that our momentum-based diffusion process achieves faster convergence w.r.t the expectation.

---

> ### Author Response · Authors · 2023-11-21
>
> Dear Reviewer GwvH,
>
> Thank you for your insightful and valuable comments, which have been of great help to us. The deadline for our discussion is approaching. If you have any further concerns or questions, welcome to discuss them with us.
>
> Moreover, another two reviewers have positive comments on this work. So could you also read their comments and decide whether you change the score? Many thanks.
>
> Bests, Authors

---

> ### Author Response · Authors · 2023-11-23
>
> Dear Reviewer GwvH:
>
> Thank you for your insightful and valuable comments, which have been of great help to us!
>
> As this is the last day of the discussion period, we would be grateful for any further thoughts or an update to the rating if you feel it would be appropriate. We are here to answer any additional questions that you may have.
>
> Thank you again! We appreciate you taking the time to review our paper.
>
> Best,
> Authors

---

### Official Review · Reviewer_DPH4 · 2023-11-10

**Soundness:** 3 good
**Presentation:** 3 good
**Contribution:** 3 good
**Rating:** 6
**Confidence:** 3

**Summary:**

This paper is trying to reduce the high computational cost of training diffusion models. The authors propose a new method called the Fast Diffusion Models (FDM), which are intuitively similar to doing momentum on SGD in stochastic optimization. FDM significantly reduces the training cost, as well as the sampling cost of DMs, while maintaining or improving their image synthesis performace. Moreover, the FDM framework is general and flexible, can be adapted to several DM frameworks including VP/VE SDE, EDM. The authors verify by experiments that the performance of FDM outperforms the corresponding baseline models under most settings.

**Strengths:**

The algorithm framework presented in this paper is both elegantly designed and robust in terms of performance. In its comparison of the score network of different diffusion models in Table 1, their method can be summarized as modifying the expectation of the perturbation kernel to incorporate a momentum related term $e^{-\int_0^{t^{\prime}} \beta(s) \mathrm{d} s}\left(1+\int_0^{t^{\prime}} \beta(s) \mathrm{d} s\right)$. The modification is simple, possibly adaptive to more diffusion models. Additionally, it has demonstrated superior performance in benchmark tests.

**Weaknesses:**

The idea of aligning diffusion process with stochastic gradient descent, and adopt acceleration techniques in SGD is not entirely new.

In [1], they first proposed the critically-damped Langevin diffusion, in which they mentioned that "the velocity in CLD accelerates mixing in the diffusion process, and is equivalent to momentum in gradient descent", where they refer to [2] for the equivalence.

I understand that removing the velocity term stabilizes training and might result in better performance compared with CLD, but I wonder how to compare the understanding in [2] that critically damped Langevin $=$ gradient descent on probability space, with the viewpoint proposed in this paper.

[1] Dockhorn, Tim, Arash Vahdat, and Karsten Kreis. "Score-based generative modeling with critically-damped langevin diffusion." arXiv preprint arXiv:2112.07068

[2] Ma, Yi-An, et al. "Is There an Analog of Nesterov Acceleration for MCMC? arXiv e-prints, page." arXiv preprint arXiv:1902.00996

**Questions:**

- The formulation of acceleration seems to be closely related to that in the critically damped Langevin diffusion. Specifically, (15) and (19) in this paper, which corresponds to the update equation of $x$ and its velocity, corresponds to (28) in Dockhorn et al. (2021) (with $v_0 = 0$). It seems that the techniques employed in EDM, particularly the conversion of the discrete process into a continuous one, play a significant role in enhancing the efficiency of the proposed method. Can the authors elaborate more on what is most critical in the superiority of their experimental results?

- Why does the mixing speed of the forward diffusion process, transitioning from the data distribution to a Gaussian distribution, relate to the training cost?

---

> ### Author Response · Authors · 2023-11-20
>
> Thank you for your insightful and positive comments! Below, we provide a point-by-point response to address your concerns. We welcome further discussion to enhance the clarity and effectiveness of our work.
>
> **Question 1: The idea of aligning diffusion process with stochastic gradient descent, and adopt acceleration techniques in SGD is not entirely new. In [1], they first proposed the critically-damped Langevin diffusion, in which they mentioned that "the velocity in CLD accelerates mixing in the diffusion process, and is equivalent to momentum in gradient descent", where they refer to [2] for the equivalence. I understand that removing the velocity term stabilizes training and might result in better performance compared with CLD, but I wonder how to compare the understanding in [2] that critically damped Langevin = gradient descent on probability space, with the viewpoint proposed in this paper.**
>
> **Response 1:**
> In their paper [2], Ma et al. first formulated gradient-based MCMC sampling as optimization in the probability space, utilizing various mathematical notations and tools such as Lebesgue measure, second moment, coupling, Wasserstein-2 distance, and optimal transport. Then they showed that an underdamped Langevin algorithm performs accelerated gradient descent by transforming accelerated gradient descent into its continuous dynamic. The most significant contribution in [2] is that they further proved the faster convergence rate of underdamped Langevin algorithm than the classical Langevin algorithm under some conditions, such as log-Sobolev inequality. These results are rigorously presented using both sample and velocity spaces, offering novel and profound insights into understanding gradient-based MCMC sampling and its accelerated version.
>
> By comparison, there are two main differences between this work and [2] because of their different focuses and targeted readers.
>
> **(1)** Our new simple and intuitive understanding complements the previous deep mathematical insights of [2] and thus allowed us to develop practically efficient algorithms. This work constructs a dynamic function directly connecting the gradient descent with the diffusion forward process. It does not borrow complex mathematical notations and tools in [2]. Then, motivated by the faster convergence speed of momentum gradient descent, this work introduces momentum to accelerate diffusion forward process, and shows the faster speed. This work focuses on more practical and efficient diffusion model design for practical designers and engineers, in contrast to the deep mathematical insights targeted in [2].
>
> **(2)** This work prioritizes efficiency, an aspect not extensively considered in both [2] and CLD [1]. This work eliminates the velocity term, yielding three benefits. **(a)** It stabilizes training, since using velocity and sample spaces like CLD and [2] needs numerically unstable Cholesky factorization (see CLD), while removing velocity allows to simplify the perturbation kernel and avoids such instability. **(b)** It reduces model memory cost which is important in practice, since using two spaces needs roughly $2\times$ model parameter and memory cost than using a single one, and poses challenges to integrate with SoTA diffusion models which are often of large model sizes. **(c)** It improves compatibility by allowing our diffusion process to be directly integrated into popular diffusion models like EDM and numerical solvers like dpm-solver, which primarily use the sample space. This is in contrast to using two spaces, which would necessitate substantial revisions to their formulations or might be incompatible.
>
> In the final version which should allow us to use an extra page, we will discuss these related works for their big efforts on understanding gradient-based MCMC sampling and its acceleration version in detail. We do not discuss this now, since there is really no space to add new content.

---

> ### Author Response · Authors · 2023-11-20
>
> **Question 2-1: The formulation of acceleration seems to be closely related to that in the critically damped Langevin diffusion. Specifically, (15) and (19) in this paper, which corresponds to the update equation of and its velocity, corresponds to (28) in Dockhorn et al. (2021) (with $v_0 = 0$).**
>
> **Response 2-1:**
> **Despite some similarity, our FDM differs CLD from significant differences.**
> When $v_0 = 0$, the velocity in CLD becomes $v_{\text{CLD}}(t) = -\mathcal{B}(t)\exp(-2\mathcal{B}(t)\Gamma^{-1}) x_0$ which differs from our FDM whose velocity is $\dot{x}_{\text{FDM}}(t) = -\beta(t)\mathcal{B}(t)\exp(-\mathcal{B}(t)) x_0$. These differences  are not caused by manual design for a superior velocity formulation, but stem from our distinct handling approach to the forward diffusion process.
>
> **(a) Velocity Diffusion:** CLD explicitly diffuses the velocity in an additional velocity space. In contrast, our FDM implicitly determines velocity through the time derivative of the sample $x_t$, which gets rid of  explicit velocity computation.
>
> **(b) Critically-Damped State:** In CLD, the critically-damped state aims to modify the noise injection level, seeking a balance between the Hamiltonian component and the Ornstein-Uhlenbeck process. However, in our FDM, the critically-damped state is designed to avoid oscillation in the expectation diffusion trajectory to streamline the model training and avoid overshoot during sampling.
>
> **Question 2-2: It seems that the techniques employed in EDM, particularly the conversion of the discrete process into a continuous one, play a significant role in enhancing the efficiency of the proposed method. Can the authors elaborate more on what is most critical in the superiority of their experimental results?**
>
> **Response 2-2:**
> **The superiority performance of our FDM is primarily based on our streamlined perturbation kernel and its wide applicability.**
>
> **(1) Streamlined Perturbation Kernel:** Our approach is grounded in the EDM design space, where we follow the spirit of treating marginal distributions as first-class citizens and design our diffusion process directly based on mean scaling $\mu_t$ and noise schedule $\sigma_t$ within marginal distributions. This design allows our momentum-based diffusion process to employ a more streamlined perturbation kernel, as shown in Eq. (14) of our paper. This contrasts with the more complex formulation of the perturbation kernel in the CLD paper (Eq. (28) to Eq. (33)), making our approach more efficient to compute and more robust against numerical errors in practical applications.
>
> **(2) Wide Applicability:** Our FDM operates solely within the sample space, with velocity implicitly determined through the time derivative of the sample $x_t$. This innovative approach enables seamless integration with advanced frameworks and numerical solvers. This compatibility not only ensures smooth integration but also has the potential to significantly enhance the performance of our model by utilizing the inherent advantages of these advanced techniques. The benefit of this approach is evident in our experimental study presented in Table 7. In the experiments, both CLD and our FDM were applied using the traditional RK45 solver. However, a significant distinction was observed when a more efficient EDM sampler was employed. While CLD exhibited a notable performance decrease with the EDM sampler, our FDM maintained consistent performance. Additionally, FDM is able to improve its efficiency by leveraging the advantages of the EDM sampler, showcasing its adaptability and effectiveness in various computational environments.
>
> Moreover, we have extended our experimental analysis to include gDDIM[3], which addresses oscillation/overshoot and numerical instability issues commonly found in the vanilla CLD. The results of these experiments are presented below. Despite the specific enhancements implemented in gDDIM to improve upon CLD, the experimental results demonstrate that our FDM still significantly outperforms gDDIM. This further underscores the robustness and efficiency of our FDM in comparison to previous methods.
>
> **Table: Image synthesis performance on CIFAR-10 with 35 NFEs. gDDIM (CLD) use specifically designed gDDIM sampler, while VP-FDM use EDM sampler. The result denoted with $\dagger$ is from the original paper, with 50 NFEs.**
> |    Method   | 30 Mimg | 50 Mimg | 70 Mimg | 100 Mimg |      Best      |
> |:-----------:|:-------:|:-------:|:-------:|:--------:|:--------------:|
> | gDDIM (CLD) |   5.15  |   2.95  |   2.76  |   2.51   | 2.25$^\dagger$ |
> | VP-FDM      |   4.01  |   2.74  |   2.37  |   2.24   |      2.08      |
>
> [3] Zhang, Qinsheng, Molei Tao, and Yongxin Chen. gDDIM: Generalized Denoising Diffusion Implicit Models. ICLR 2023.

---

> ### Author Response · Authors · 2023-11-20
>
> **Question 3: Why does the mixing speed of the forward diffusion process, transitioning from the data distribution to a Gaussian distribution, relate to the training cost?**
>
> **Response 3:**
> **The convergence speed of the forward diffusion process is closely related to the training cost, because it directly determines the score function, while matching this score function is the training objective of the diffusion model.** If the forward process is too fast, then the reverse process is pushed to be fast and enforces the network to predict a too rapidly changed trajectory, which increases the learning challenges of the network; otherwise, the forward process is too slow, the trajectory converges to a Gaussian noise too slowly and hence slows down the learning of network.
>
> Fortunately, momentum can well trade off the convergence speed and the learning difficulty of the trajectory. During the early stages of diffusion (at a small $t$), the momentum in FDM accumulates historical information, thereby increasing the speed of the forward process as the model gains confidence in identifying the direction indicated by the score function. This implies that the score function at this stage is relatively easy to match. In later stages (at a large $t$), when the forward process nears convergence to the Gaussian distribution, the momentum gradually decreases, thus slowing down the forward process as it becomes more challenging to identify the direction due to the increasing noise. This indicates that the score function at this stage becomes more difficult to match.
> This adaptive adjustment in "step size" in FDM leads to more effective navigation of the diffusion trajectory, which optimizes the balance between the speed of convergence to the target Gaussian distribution and the learning challenge posed to the network. The efficacy of this approach is demonstrated by our empirical results based on VP, VE, and EDM.

---

> ### Author Response · Authors · 2023-11-21
>
> Dear Reviewer DPH4,
>
> Thank you for your insightful and positive comments, which have been of great help to us. The deadline for our discussion is approaching. If you have any further concerns or questions, welcome to discuss them with us.
>
> Bests, Authors

---

> ### Author Response · Authors · 2023-11-23
>
> Dear Reviewer DPH4:
>
> Thank you for your insightful and positive comments, which have been of great help to us.
>
> As this is the last day of the discussion period, we would be grateful for any further thoughts or an update to the rating if you feel it would be appropriate. We are here to answer any additional questions that you may have.
>
> Thank you again! We appreciate you taking the time to review our paper.
>
> Best, Authors

---

### Meta-Review · Program_Chairs · 2023-12-07

**Metareview:**

Program chair note: The program chairs looked closely in this paper, its reviews and its meta-review. Bringing up an arxiv paper that has been published within 4 months earlier to the ICLR submission deadline violates the ICLR guidelines and cannot be used for justification of lack of novelty. The rest of the meta-review stands as is.

The paper proposes an approach that incorporates momentum to diffusion models by mapping diffusion models and momentum-based stochastic gradient descent.

The paper explores an interesting connection and the mix of theory and practice is promising.  The author feedback satisfactorily addressed many of the reviewers' points.

Unfortunately, however, significant concerns remain on the 'looseness' of the connection between SGD and DMs, and the gap between theory and practice in terms of training.

The connection between SGD and DDPM presented in section 4.1 is problematic as the function f one is trying to minimized itself depends on t, while the iterates also depend on t.

There is a serious disconnect between theory and claims. Indeed the theoretical results concern the forward process and the authors state that the benefits transfer to the reverse process "since the reverse process is decided by its forward diffusion process". Such a statement if far from sufficient to explain what faster training should be expected.

While it is ok to decouple mean and variance, the incorporation of the variance into the perturbation Kernel in equation (14) amounts to ignoring the fact that the momentum would actually also impact the variance and not merely the mean. This restriction should be discussed.

In addition, as noted by several reviewers, the novelty of the work is somewhat limited given prior work leveraging momentum for DMs (see for instance "Diffusion Sampling with Momentum for Mitigating Divergence Artifacts" (https://arxiv.org/abs/2307.11118).

We strongly encourage the authors to revise their work taking the above into account to strengthen their analysis and how it translates to practice.

**Justification For Why Not Higher Score:**

The paper is problematic from a mathematical standpoint, and also some claims are unsubstantiated.

**Justification For Why Not Lower Score:**

N/A

---

### Decision · Program_Chairs · 2024-01-16

Reject